# 🍌 PAPERBANANA: Automating Academic Illustration for AI Scientists

**Dawei Zhu** [1][2] **Rui Meng** [2] **Yale Song** [2] **Xiyu Wei** [1] **Sujian Li** [1] **Tomas Pfister** [2] **Jinsung Yoon** [2]

https://github.com/dwzhu-pku/PaperBanana

## Abstract

Despite rapid advances in autonomous AI scientists powered by language models, generating publication-ready illustrations remains a labor-intensive bottleneck in the research workflow. To lift this burden, we introduce PAPERBANANA, an agentic framework for automated generation of publication-ready academic illustrations. Powered by state-of-the-art VLMs and image generation models, PAPERBANANA orchestrates specialized agents to retrieve references, plan content and style, render images, and iteratively refine via self-critique. To rigorously evaluate our framework, we introduce PAPERBANANABENCH, comprising 292 test cases for methodology diagrams curated from NeurIPS 2025 publications, covering diverse research domains and illustration styles. Comprehensive experiments demonstrate that PAPERBANANA consistently outperforms leading baselines in faithfulness, conciseness, readability, and aesthetics. We further show that our method effectively extends to the generation of high-quality statistical plots. Collectively, PAPERBANANA paves the way for the automated generation of publication-ready illustrations.

## 1. Introduction

Autonomous scientific discovery is a long-standing pursuit of artificial general intelligence (Langley et al., 1987; Langley, 2024; Schmidhuber, 2010; Ghahramani, 2015). With the rapid evolution of Large Language Models (LLMs) (Comanici et al., 2025; Anthropic, 2025; OpenAI, 2025b; Liu et al., 2024; Yang et al., 2025a), *autonomous AI Scientists*

---

This work was done while Dawei was a student researcher at Google Cloud AI Research. [1]School of Computer Science, Peking University [2]Google Cloud AI Research. Correspondence to: Jinsung Yoon <jinsungyoon@google.com>, Sujian Li <lisujian@pku.edu.cn>, Dawei Zhu <dwzhu@pku.edu.cn>.

have demonstrated the potential to automate many facets of the research lifecycle, such as literature review, idea generation, and experiment iteration (Lu et al., 2024; Luo et al., 2025; Gottweis et al., 2025). Yet scientific discoveries achieve their full value only through effective communication. Despite their proficiency in textual analysis and code execution, current autonomous AI scientists struggle to visually communicate discoveries, especially for generating illustrations (diagrams and plots) that adhere to the rigorous standards of academic manuscripts.

Among these illustration tasks, generating methodology diagrams represents a significant challenge, demanding both content fidelity and visual aesthetics. Prior endeavors for diagram generation have predominantly adopted the code-based paradigm, leveraging TikZ (Belouadi & Eger, 2024; Belouadi et al., 2025), Python-PPTX (Zheng et al., 2025), or SVG to programmatically synthesize diagrams. While effective for structured content, these methods can encounter expressiveness limitations when attempting to produce the intricate visual elements – such as specialized icons and custom shapes – that are increasingly common in modern AI publications. Conversely, although recent image generation models (Deepmind, 2025; OpenAI, 2025a) have demonstrated advanced instruction-following capabilities and high-quality visual outputs, consistently generating academic illustrations that meet scholarly standards remains a difficult task (Zuo et al., 2025). Specialized expertise required for professional illustration tools often constrains researchers' ability to freely express complex ideas, forcing them to invest substantial manual effort into crafting figures. This creates a significant bottleneck in the effective visual communication of scientific discoveries.

In this paper, we introduce PAPERBANANA, an agentic framework designed to bridge this gap by automating the production of high-quality academic illustrations. Given a methodology description and diagram caption as input, PAPERBANANA orchestrates specialized agents powered by state-of-the-art VLMs and image generation models (e.g. Gemini-3-Pro and Nano-Banana-Pro) to retrieve reference examples, devise detailed plans for content and style, render images, and iteratively refine via self-critique. This reference-driven collaborative workflow allows the system

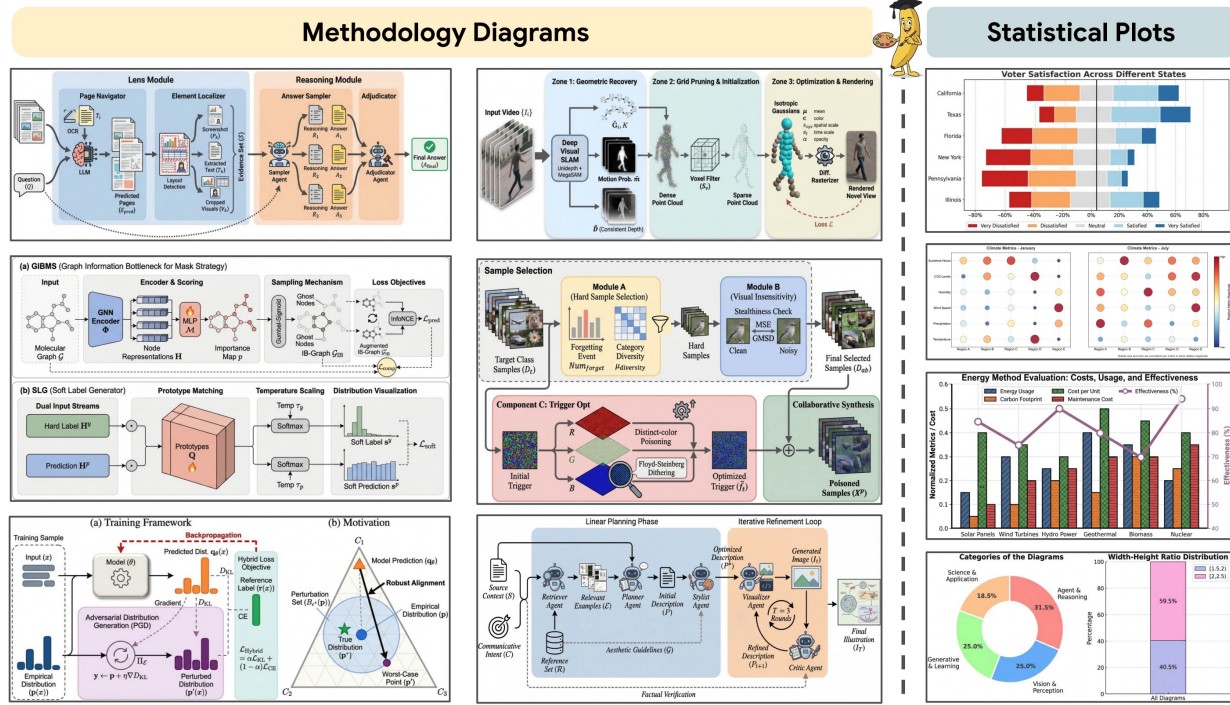

*Figure 1.* Examples of methodology diagrams and statistical plots generated by PAPERBANANA, which show the potential of automating the generation of academic illustrations.

to effectively master the logical composition and stylistic norms required for publication-ready illustrations. Beyond methodology diagrams, our framework demonstrates significant versatility by extending to statistical plots, offering a comprehensive solution for scientific visualization.

To rigorously evaluate our framework and address the absence of dedicated benchmarks for automated academic illustration, we introduce PAPERBANANABENCH, a comprehensive benchmark for methodology diagram generation. The benchmark comprises 292 test cases and 292 reference cases curated from NeurIPS 2025 publications, spanning diverse research topics and illustration styles. To assess generation quality, we employ a VLM-as-a-Judge approach for reference-based scoring against human illustrations across four dimensions: faithfulness, conciseness, readability, and aesthetics, with reliability verified through correlation with human judgments.

Comprehensive experiments on our benchmark demonstrate the effectiveness of PAPERBANANA. Our method consistently outperforms leading baselines across all four evaluation dimensions—faithfulness (+2.8%), conciseness (+37.2%), readability (+12.9%), and aesthetics (+6.6%)—as well as the aggregated overall score (+17.0%) for diagram generation. We further show that our method also seamlessly extends to statistical plots. Collectively, our method paves the way for automating the generation of academic illustrations (Examples shown in Figure 1). As a

demonstration of its capability, figures marked with 🍌 in this manuscript were entirely generated using PAPERBANANA. Additionally, we discuss intriguing settings including using our framework to enhance existing human-created illustrations and using image generation models for statistical plot generation. To sum up, our contributions are:

- We propose PAPERBANANA, a fully automated agentic framework that orchestrates specialized agents to generate publication-ready academic illustrations.

- We construct PAPERBANANABENCH to assess the quality of academic illustrations, particularly methodology diagrams.

- Comprehensive experiments show that our workflow significantly outperforms leading baselines, showing promise for automating the generation of academic illustrations.

**Conflict of Interest Disclosure.** The authors R.M., Y.S., T.P., and J.Y. are employed by Google, which develops Gemini-3-Pro and Nano-Banana-Pro, models used and evaluated in this paper as components of the proposed framework and as baseline or judge models in the experiments.

## 2. Task Formulation

We formalize the task of automated academic illustration generation as learning a mapping from a source context and

## PaperBanana Framework

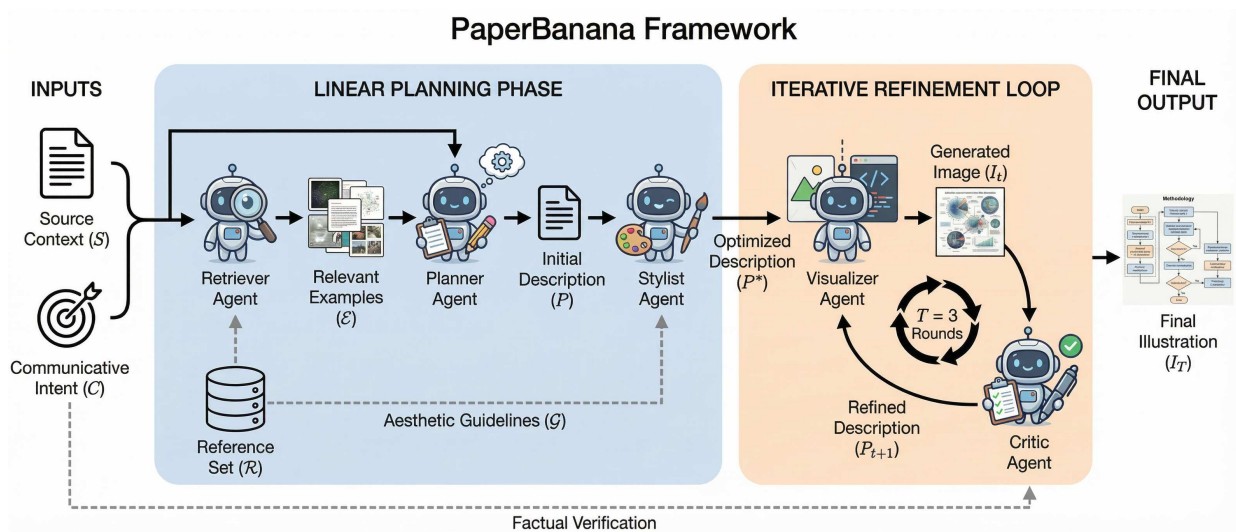

*Figure 2.* [Generated by 🐌, textual description to reproduce this diagram is presented in Appendix F.] Overview of our PAPERBANANA framework. Given the source context and communicative intent, we first apply a *Linear Planning Phase* to retrieve relevant reference examples and synthesize a stylistically optimized description. We then use an *Iterative Refinement Loop* (consisting of *Visualizer* and *Critic Agents*) to transform the description into visual output and conduct multi-round refinements to produce the final academic illustration.

a communicative intent to a visual representation. Let $S$ denote the source context containing the essential information, and $C$ denote the communicative intent that specifies the scope and focus of the desired illustration. The goal is to generate an image $I$ that faithfully visualizes $S$ while fulfilling the communicative intent $C$, formulated as:

$$I = f(S, C). \tag{1}$$

To further guide the mapping function, the input can be optionally augmented by a set of $N$ reference examples $\mathcal{E} = \{E_n\}_{n=1}^N$. Each example $E_n$ serves as a ground-truth demonstration, defined as a tuple $E_n = (S_n, C_n, I_n)$, where $I_n$ is the reference illustration corresponding to the context $S_n$ and communicative intent $C_n$. Integrating this, the unified task formulation becomes:

$$I = f(S, C, \mathcal{E}), \tag{2}$$

where $\mathcal{E}$ defaults to $\emptyset$ when no examples are used (i.e., zero-shot generation).

Among various types of academic illustrations, this paper primarily focuses on the automated generation of methodology diagrams, which requires interpreting complex technical concepts and logical flows from textual descriptions into high-fidelity, visually pleasing illustrations. In this setting, the source context $S$ is the textual description of the method (e.g., methodology sections), and the communicative intent $C$ is the figure caption specifying the scope and focus (e.g., "Overview of our framework").

## 3. Methodology

In this section, we present the architecture of PAPERBANANA, a reference-driven agentic framework for automated academic illustration. As illustrated in Figure 2, PAPERBANANA orchestrates a collaborative team of five specialized agents—Retriever, Planner, Stylist, Visualizer, and Critic—to transform raw scientific content into publication-quality diagrams and plots. (See Appendix H for prompts)

**Retriever Agent.** Given the source context $S$ and the communicative intent $C$, the Retriever Agent identifies $N$ most relevant examples $\mathcal{E} = \{E_n\}_{n=1}^N \subset \mathcal{R}$ from the fixed reference set $\mathcal{R}$ to guide the downstream agents. As defined in Section 2, each example $E_i \in \mathcal{R}$ is a triplet $(S_i, C_i, I_i)$. To leverage the reasoning capabilities of VLMs, we adopt a generative retrieval approach where the VLM performs selection over candidate metadata:

$$\mathcal{E} = \text{VLM}_{\text{Ret}}\left(S, C, \{(S_i, C_i)\}_{E_i \in \mathcal{R}}\right) \tag{3}$$

Specifically, the VLM is instructed to rank candidates by matching both research domain (e.g., Agent & Reasoning) and diagram type (e.g., pipeline, architecture), with visual structure being prioritized over topic similarity. By explicitly reasoned selection of reference illustrations $I_i$ whose corresponding contexts $(S_i, C_i)$ best match the current requirements, the Retriever provides a concrete foundation for both structural logic and visual style.

**Planner Agent.** The Planner Agent serves as the cognitive core of the system. It takes the source context $S$, communicative intent $C$, and retrieved examples $\mathcal{E}$ as inputs. By performing in-context learning from the demonstrations in

$\mathcal{E}$, the Planner translates the unstructured or structured data in $S$ into a comprehensive and detailed textual description $P$ of the target illustration:

$$P = \text{VLM}_{\text{plan}}(S, C, \{(S_i, C_i, I_i)\}_{E_i \in \mathcal{E}}). \quad (4)$$

**Stylist Agent.** To ensure the output adheres to the aesthetic standards of modern academic manuscripts, the Stylist Agent acts as a design consultant. A primary challenge lies in defining a comprehensive "academic style," as manual definitions are often incomplete. To address this, the Stylist traverses the entire reference collection $\mathcal{R}$ to automatically synthesize an *Aesthetic Guideline* $\mathcal{G}$ covering key dimensions such as color palette, shapes and containers, lines and arrows, layout and composition, and typography and icons (see Appendix G for the summarized guideline and implementation details). Armed with this guideline, the Stylist refines each initial description $P$ into a stylistically optimized version $P^*$:

$$P^* = \text{VLM}_{\text{style}}(P, \mathcal{G}). \quad (5)$$

This ensures that the final illustration is not only accurate but also visually professional.

**Visualizer Agent.** After receiving the stylistically optimized description $P^*$, the Visualizer Agent collaborates with the Critic Agent to render academic illustrations and iteratively refine their quality. The Visualizer Agent leverages an image generation model to transform textual descriptions into visual output. In each iteration $t$, given a description $P_t$, the Visualizer generates:

$$I_t = \text{Image-Gen}(P_t), \quad (6)$$

where the initial description $P_0$ is set to $P^*$.

**Critic Agent.** The Critic Agent forms a closed-loop refinement mechanism with the Visualizer by closely examining the generated image $I_t$ and providing refined description $P_{t+1}$ to the Visualizer. Upon receiving the generated image $I_t$ at iteration $t$, the Critic inspects it against the original source context $(S, C)$ to identify factual misalignments, visual glitches, or areas for improvement. It then provides targeted feedback and produces a refined description $P_{t+1}$ that addresses the identified issues:

$$P_{t+1} = \text{VLM}_{\text{critic}}(I_t, S, C, P_t). \quad (7)$$

This revised description is then fed back to the Visualizer for regeneration. The Visualizer-Critic loop iterates for $T = 3$ rounds, with the final output being $I = I_T$. This iterative refinement process ensures that the final illustration meets the high standards required for academic dissemination.

**Extension to Statistical Plots.** The framework extends to statistical plots by adjusting the Visualizer and Critic

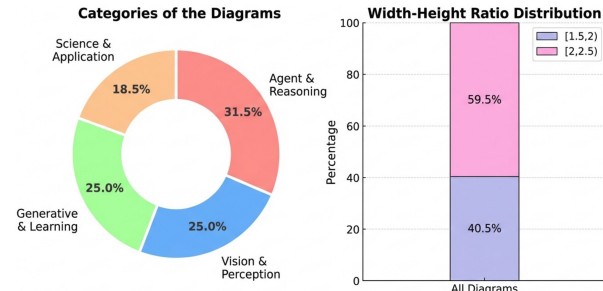

*Figure 3.* [Generated by 🍌] Statistics of the test set of PAPER-BANANABENCH (totaling 292 samples). The average length of source context is 3,020.1 words, while the average length of figure caption is 70.4 words.

agents. For numerical precision, the Visualizer converts the description $P_t$ into executable Python Matplotlib code: $I_t = \text{VLM}_{\text{code}}(P_t)$. The Critic evaluates the rendered plot and generates a refined description $P_{t+1}$ addressing inaccuracies or imperfections: $P_{t+1} = \text{VLM}_{\text{critic}}(I_t, S, C, P_t)$. The same $T = 3$ round iterative refinement process applies. While we prioritize this code-based approach for accuracy, we also explore direct image generation in Section 6. See Appendix H.2 for adjusted prompts.

## 4. Benchmark Construction

The lack of benchmarks hinders rigorous evaluation of automated diagram generation. We address this with PA-PERBANANABENCH, a dedicated benchmark curated from NeurIPS 2025 methodology diagrams, capturing the sophisticated aesthetics and diverse logical compositions of modern AI papers. We detail the construction pipeline and evaluation protocol below; dataset statistics are in Figure 3.

### 4.1. Data Curation

**Collection & Parsing.** We begin by randomly sampling 2,000 papers from the 5,275 publications at NeurIPS 2025 and retrieving their PDF files. Subsequently, we utilize the MinerU toolkit (Niu et al., 2025) to parse these documents, extracting the text of the methodology sections, and all the diagrams and their captions in the paper.

**Filtering.** We then apply a filtering stage to ensure data quality. First, we discard papers without methodology diagrams, yielding 1,359 valid candidates. Second, we restrict the aspect ratio ($w : h$) to $[1.5, 2.5]$. Ratios below 1.5 are excluded as methodology diagrams typically require wider landscape layouts for logical flows, while ratios exceeding 2.5 are unsupported by current image generation models. Including such outliers would introduce bias in side-by-side evaluations by revealing the human origin of candidates. This yields 610 valid candidates, each a tuple $(S, I, C)$, where $S$ is the methodology description, $I$ is the methodol-

ogy diagram, and $C$ is the caption.

**Categorization.** To facilitate future analysis of generating different types of diagrams, we further categorize the diagrams into four classes, based on visual topology and content: *Agent & Reasoning*, *Vision & Perception*, *Generative & Learning*, and *Science & Applications* (see Appendix D for definitions). Gemini-3-Pro is used to perform the categorization, assigning samples with hybrid elements to their predominant category.

**Human Curation.** Finally, we conduct a human curation phase to guarantee the integrity and quality of the dataset. Annotators are tasked with verifying and correcting the extracted methodology descriptions and captions, validating the correctness of diagram categorizations, and filtering out diagrams of insufficient visual quality (e.g., overly simplistic, cluttered, or abstract designs). Following this rigorous process, 584 valid samples remain. We randomly partition these into two equal subsets: a test set ($N = 292$) for evaluation and a reference set ($N = 292$) to facilitate retrieval-augmented in-context learning.

### 4.2. Evaluation Protocol

We utilize VLM-as-a-Judge to assess the quality of methodology diagrams and statistical plots. Given the inherent subjectivity in evaluating visual design, we employ a referenced comparison approach where the judge compares the model-generated diagram against the human-drawn diagram to determine which better satisfies each evaluation criterion.

**Evaluation Dimensions.** Inspired by Quispel et al. (2018), we evaluate diagrams on two perspectives. Detailed rubrics for each dimension are provided in Appendix I.

- **Content (Faithfulness & Conciseness):** *Faithfulness* ensures alignment with the source context (methodology description) and communicative intent (caption), while *Conciseness* requires focusing on core information without visual clutter.
- **Presentation (Readability & Aesthetics):** *Readability* demands intelligible layouts, legible text, no excessive crossing lines, etc. *Aesthetics* evaluates adherence to the stylistic norms of academic manuscripts.

**Referenced Scoring.** For each dimension, the VLM judge compares the model-generated diagram against the human reference given the context and caption. It determines *Model wins*, *Human wins*, or *Tie* based on relative quality, which are then mapped to scores of 100, 0, and 50, respectively. To aggregate scores into an overall metric, we follow the design principle that information visualization must primarily "show the truth" (Tufte, 1983; Mackinlay, 1986; Quispel et al., 2018). We employ a hierarchical aggregation strategy, designating faithfulness and readability as **primary**

dimensions, and conciseness and aesthetics as **secondary**. If primary dimensions yield a decisive winner (i.e., winning both, or winning one with a tie), this determines the overall winner. In case of a tie (e.g., each wins one, or both tie), we apply the same rule to the secondary dimensions. This hierarchical approach ensures that content fidelity and clarity take precedence over aesthetics and conciseness.

## 5. Experiments

### 5.1. Baseline Methods and Models

We compare PAPERBANANA against four baseline settings: (1) *Vanilla*, directly prompting the image generation model to generate diagrams based on the input context (methodology description and caption); (2) *Few-shot*, building upon the vanilla baseline by augmenting the prompt with 10 few-shot examples, where each example consists of a triplet (methodology description, caption, diagram) to enable in-context learning for the image generation model; (3) *Paper2Any* (Liu et al., 2025), an agentic framework that generates diagrams to present high-level ideas of the papers, which is the closest to our setting; and (4) *AutoFigure*[†] (Zhu et al., 2026), a concurrent system for generating publication-ready scientific illustrations from paper content. For VLM backbone, we default to Gemini-3-Pro, while for image generation model, we experiment with Nano-Banana-Pro and GPT-Image-1.5. (See Appendix D for more implementation details.)

### 5.2. Evaluation Settings.

Evaluating the quality of generated diagrams demands strong visual perception and understanding capabilities, particularly for the Faithfulness dimension, which requires accurately identifying and interpreting subtle modules and connections. Hence, we employ Gemini-3-Pro as our VLM-based Judge. To validate its reliability, we randomly sampled 50 cases (25 from vanilla and 25 from our method) and conducted a two-fold validation process:

**Inter-Model Agreement (Consistency).** First, we verify that our evaluation protocol is robust and model-agnostic. We evaluated the agreement between our judge (Gemini-3-Pro) and other distinct VLMs (Gemini-3-Flash and GPT-5). Kendall's tau correlations with Gemini-3-Flash across the four dimensions (Faithfulness, Conciseness, Readability, Aesthetics) and their aggregation are 0.51, 0.60, 0.45, 0.56, and 0.55, respectively; correlations with GPT-5 are 0.43, 0.47, 0.44, 0.42, and 0.45, respectively. This confirms the consistency of our protocol across different judge models[1].

---

[1]According to existing literature (Hollander et al., 2013; Cohen, 2013), a Kendall's tau correlation exceeding 0.4 is generally considered to represent relatively strong agreement.

*Table 1.* Main results on PAPERBANANABENCH. Best score in each column is in **bold**. [†] denotes evaluation on a 49-case subset.

| Method | Faithfulness ↑ | Conciseness ↑ | Readability ↑ | Aesthetic ↑ | Overall ↑ |
|---|---|---|---|---|---|
| *Vanilla Settings* | | | | | |
| GPT-Image-1.5 | 4.5 | 37.5 | 30.0 | 37.0 | 11.5 |
| Nano-Banana-Pro | 43.0 | 43.5 | 38.5 | 65.5 | 43.2 |
| Few-shot Nano-Banana-Pro | 41.6 | 49.6 | 37.6 | 60.5 | 41.8 |
| *Agentic Frameworks* | | | | | |
| **Paper2Any** (w/ Nano-Banana-Pro) | 6.5 | 44.0 | 20.5 | 40.0 | 8.5 |
| **AutoFigure**[†] | 37.0 | 10.2 | 10.0 | 14.0 | 8.3 |
| **PAPERBANANA (Ours)** | | | | | |
|    w/ GPT-Image-1.5 | 16.0 | 65.0 | 33.0 | 56.0 | 19.0 |
|    w/ Nano-Banana-Pro | 45.8 | **80.7** | **51.4** | **72.1** | **60.2** |
| Human | **50.0** | 50.0 | 50.0 | 50.0 | 50.0 |

**Human Alignment (Validity).** Second, we verify that our VLM judge is a valid proxy for human evaluation. We tasked two human annotators to independently perform reference-based scoring on the same 50 samples using the same rubrics, followed by a discussion to reach consensus on conflicting cases. Kendall's tau correlations between Gemini-3-Pro and human annotations are 0.43, 0.57, 0.45, 0.41, and 0.45, respectively. These strong correlations demonstrate that our VLM-based judge aligns well with human perception. (See Appendix C for more details.)

### 5.3. Main Results

Table 1 summarizes the performance of our method and baseline methods on PAPERBANANABENCH. PAPER-BANANA consistently outperforms leading baselines across all metrics. We attribute the poor performance of GPT-Image in both vanilla and agentic settings to its weaker instruction following and text rendering capabilities compared to Nano-Banana-Pro, which fails to meet the strict requirements of academic illustration. Similarly, while Paper2Any also supports generating paper figures, it prioritizes the presentation of high-level ideas rather than the faithful depiction of specific methodological flows necessary for methodology diagrams. AutoFigure obtains relatively higher Faithfulness than Paper2Any on the 49-case subset, but substantially lower Conciseness, Readability, and Aesthetics, indicating that retrieval-grounded style guidance and iterative refinement are important for producing polished academic methodology diagrams. These differences lead to the baselines' underperformance in our evaluation setting.

In contrast, PAPERBANANA achieves comprehensive improvements over the Vanilla Nano-Banana-Pro baseline: Faithfulness (+2.8%), Conciseness (+37.2%), Readability (+12.9%), and Aesthetics (+6.6%), contributing to a +17.0% gain in the Overall score. Regarding performance across categories, Agent & Reasoning achieves the highest over-

all score (69.9%), followed by Scientific & Application (58.8%) and Generative & Learning (57.0%), while Vision & Perception scores the lowest (52.1%). We also conducted a blind human evaluation on a subset of 50 cases to compare PAPERBANANA against vanilla Nano-Banana-Pro (see Appendix C for details). The average win / tie / loss rate of PAPERBANANA from three human judges is 72.7% / 20.7% / 6.6%, respectively. This further validates that our agentic workflow shows promising improvements in automated methodology diagram generation. See Appendix Figure 7 for case studies.

Despite the progress, we note that PAPERBANANA still underperforms the human reference in terms of faithfulness. We have included some failure analysis in Appendix Figure 10 to provide insights into the challenges.

Beyond the default full-triplet reference setting, Appendix Figure 11 provides a qualitative analysis showing that reference images alone can also provide useful stylistic and structural cues, although the full triplet remains more informative for learning the text-to-illustration mapping. We further provide qualitative cross-domain examples on biomedical papers from *Nature Communications* in Appendix Figure 12, suggesting that the framework can produce reasonable scientific illustrations beyond the machine learning methodology diagrams used to construct PAPERBANANABENCH.

### 5.4. Ablation Study

To understand the contribution of each agent component, we conduct an ablation study, with results presented in Table 2.

**Impact of the Retriever Agent.** We compare the semantic retriever with random and no-retriever baselines (rows ④–⑥ in Table 2). Without reference examples as guidance, the no-retriever setting significantly underperforms in Conciseness, Readability, and Aesthetics, as the Planner defaults to verbose, exhaustive descriptions. Moreover,

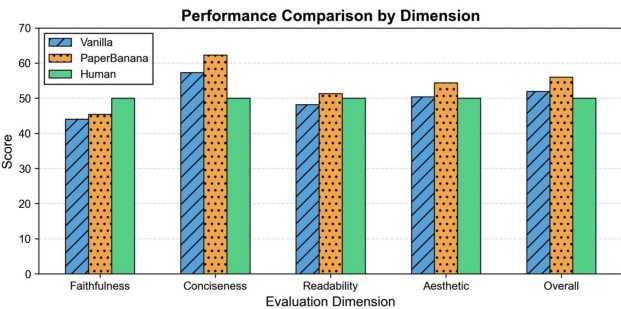

*Figure 4.* [Generated by 🍌] Vanilla Gemini-3-Pro vs. PAPER-BANANA on the statistical plots generation test set. *F*, *C*, *R*, *A* is short for *Faithfulness*, *Conciseness*, *Readability*, and *Aesthetics*, respectively.

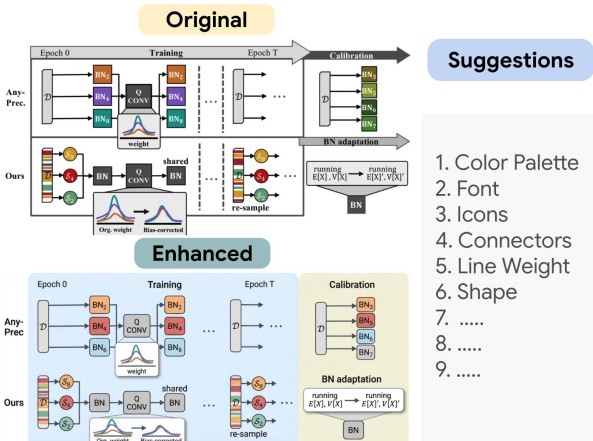

*Figure 5.* Example of enhancing aesthetics of human-drawn diagrams. The original figure is from Kim et al. (2025). More examples are provided in Figure 8.

lacking exposure to academic diagram aesthetics, this setting produces visually less refined outputs. Interestingly, the random retriever achieves performance comparable to the semantic approach, suggesting that providing general structural and stylistic patterns is more critical than precise content matching.

**Impact of the Stylist and Critic Agents.** Comparing rows ③ and ④ shows that the Stylist boosts Conciseness (+17.5%) and Aesthetics (+4.7%) but lowers Faithfulness (-8.5%), as visual polishing sometimes omits technical details. However, the Critic Agent (row ① vs. ③) effectively bridges this gap, substantially recovering Faithfulness. Additional iterations further enhance all metrics, ensuring a balance between aesthetics and technical accuracy.

## 5.5. Runtime and Cost

The iterative refinement loop introduces moderate additional overhead. For generating a single methodology diagram with Nano-Banana-Pro, the full PAPERBANANA pipeline with $T = 3$ refinement iterations costs approximately $2.5 and takes about 10 minutes on average. Removing the refinement loop reduces the cost to approximately $2.1 and the runtime to about 5 minutes. Thus, the Critic-based refinement adds roughly $0.4 and 5 minutes per figure, while consistently improving all five evaluation dimensions as shown in Table 2. We view this as a practical trade-off, especially compared with the substantially longer manual effort typically required to craft publication-quality methodology diagrams.

## 5.6. PAPERBANANA for Statistical Plots Generation.

PAPERBANANA operates by first synthesizing a detailed description of the target illustration, then visualizing it into an image. Unlike methodology diagrams that prioritize aesthetics and logical coherence, statistical plots demand rigorous numerical precision, making standard image generation models unsuitable. To address this, we demonstrate that by adopting executable code for visualization, PAPERBANANA seamlessly extends to statistical plot generation.

**Testset Curation.** Following the task formulation in Section 2, we assess PAPERBANANA's capability to generate statistical plots from tabular data and brief visual descriptions. Since raw data of statistical plots is rarely available in academic manuscripts, we repurpose ChartMimic (Yang et al., 2025b), a dataset originally constructed for chart-to-code generation. This dataset primarily includes statistical plots from arXiv papers and Matplotlib galleries, paired with human-curated Python code. Leveraging Gemini-3-Pro, we extract the underlying tabular data from the code and synthesize a brief description for each plot. Following rigorous filtering and sampling (see Appendix E), we curate 240 test cases and 240 reference examples, stratified across seven plot categories—bar chart, line chart, tree & pie chart, scatter plot, heatmap, radar chart, and miscellaneous—and two complexity levels (easy and hard). For evaluation, we adhere to the protocol detailed in Section 4, with prompts specifically tailored to statistical plots.

Figure 4 compares PAPERBANANA with vanilla Gemini-3-Pro on our curated test set. Our method consistently outperforms the baseline across all dimensions, achieving gains of +1.4%, +5.0%, +3.1%, and +4.0% in Faithfulness, Conciseness, Readability, and Aesthetics, respectively, resulting in a +4.1% overall improvement. Notably, PAPERBANANA obtains scores above the human-reference anchor in Conciseness, Readability, and Aesthetics while remaining competitive in Faithfulness. Since the human score is a pairwise scoring anchor rather than an absolute upper bound, this result should be interpreted as showing that PAPERBANANA can produce more concise and visually polished plots than the reference plots in some cases.

*Table 2.* Ablation study on PAPERBANANABENCH. The shaded row indicates the default setting of PAPERBANANA. We systematically ablate each agent component to assess its contribution. The ◯ symbol denotes the Random Retriever which randomly selects 10 examples instead of performing semantic retrieval.

| # | Module | | | | | Faithfulness ↑ | Conciseness ↑ | Readability ↑ | Aesthetic ↑ | Overall ↑ |
|---|---|---|---|---|---|---|---|---|---|---|
| | Retriever | Planner | Stylist | Visualizer | Critic | | | | | |
| ① | ✓ | ✓ | ✓ | ✓ | 3 iters | **45.8** | **80.7** | **51.4** | **72.1** | **60.2** |
| ② | ✓ | ✓ | ✓ | ✓ | 1 iter | 38.3 | 75.2 | 50.6 | 68.9 | 51.8 |
| ③ | ✓ | ✓ | ✓ | ✓ | - | 30.7 | 79.2 | 47.0 | 72.1 | 45.6 |
| ④ | ✓ | ✓ | - | ✓ | - | 39.2 | 61.7 | 47.9 | 67.4 | 49.2 |
| ⑤ | ◯ | ✓ | - | ✓ | - | 37.3 | 62.7 | 51.1 | 65.6 | 48.3 |
| ⑥ | - | ✓ | - | ✓ | - | 41.9 | 58.6 | 43.1 | 62.9 | 44.2 |

# 6. Discussion

## 6.1. Enhancing Aesthetics of Human-Drawn Diagrams

Given the summarized aesthetic guidelines $\mathcal{G}$, an intriguing question arises: can these guidelines serve to elevate the aesthetic quality of existing human-drawn diagrams? To explore this, we implement a streamlined pipeline where Gemini-3-Pro first formulates up to 10 actionable suggestions based on the original diagram and $\mathcal{G}$, which are then executed by Nano-Banana-Pro to refine the image. We evaluate the results using our reference-based protocol, comparing the refined output against the original human-drawn diagram. Across the 292 test cases, the refined diagrams achieved a win / tie / loss ratio of 56.2% / 6.8% / 37.0% in aesthetics against their original counterparts, showing that the summarized aesthetic guidelines can indeed serve to elevate the aesthetic quality of existing human-authored diagrams. An illustrative example is provided in Figure 5.

## 6.2. Coding vs Image Generation for Statistical Plots

For statistical plots, code-based approaches have demonstrated remarkable efficacy, as evidenced by Figure 4 and prior studies (Chen et al., 2025; Yang et al., 2024; Goswami et al., 2025). Given the advanced fidelity and visual appeal of recent image generation models, we compare code-based (Gemini-3-Pro) and image-generation-based (Nano-Banana-Pro) approaches for the Visualizer agent in PAPERBANANA, as shown in Figure 6. Results reveal distinct trade-offs: image generation excels in presentation (Readability and Aesthetics) but underperforms in content fidelity (Faithfulness and Conciseness). Manual inspection shows that while image models faithfully render sparse plots, they struggle with dense or complex data, exhibiting numerical hallucinations or element repetition (Appendix Figure 9). Thus, hybridly using image generation for sparse visualizations and code for dense plots may offer the best balance.

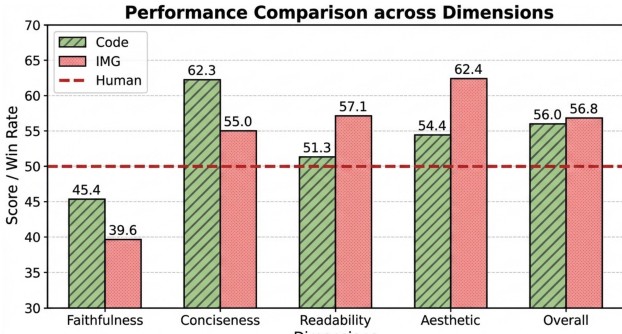

*Figure 6.* [Generated by 🍌] Coding vs. Image Generation for visualizing statistical plots.

# 7. Related Work

## 7.1. Academic Figure and Diagram Generation.

Automated academic figure generation remains a long-standing challenge (Rodriguez et al., 2023). Prior work spans several related but distinct lines.

**Vector graphics synthesis.** These methods generate diagrams through symbolic programs such as TikZ, SVG, Python-PPTX, or presentation-editing code (Belouadi et al., 2023; Belouadi & Eger, 2024; Belouadi et al., 2025; Zhang et al., 2025; Zheng et al., 2025; Pang et al., 2025). They provide strong editability and precise control over structured layouts, but can be less expressive for the diverse icons, customized shapes, and polished visual styles commonly found in modern AI methodology diagrams.

**Raster figure generation.** These approaches leverage image generation models to synthesize visually rich figures from textual or structured inputs (Deepmind, 2025; OpenAI, 2025a; Team et al., 2025; Tang et al., 2026; Zuo et al., 2025). Concurrent to our work, AutoFigure (Zhu et al., 2026) and AutoFigure-Edit (Lin et al., 2026) transform scientific content into symbolic representations before rendering figures with GPT-Image. In comparison, PAPERBANANA focuses on methodology diagrams and academic plots through a

retrieval-grounded agentic workflow: it retrieves relevant paper figures, separates content planning from style refinement, and applies iterative critique to improve both faithfulness and presentation.

**Editable reconstruction.** This line aims to recover or produce figures in formats that are easier for users to modify. Paper2Any (Liu et al., 2025) reconstructs paper content into editable visual artifacts, AutoFigure-Edit (Lin et al., 2026) targets editable scientific illustrations, and SciFig (Huang et al., 2026) explores automatic scientific figure generation with stronger emphasis on editable outputs. These systems are complementary to our goal: they emphasize post-generation editability, whereas PAPERBANANA prioritizes faithful and visually polished generation from paper content. Bridging raster generation with editable reconstruction remains an important future direction.

**Diagram and infographic benchmarks.** These works provide evaluation resources for this emerging task. Most closely related to PAPERBANANABENCH is SridBench (Chang et al., 2025), which evaluates automated diagram generation from method sections and captions across computer science and natural science domains. PAPERBANANABENCH differs by focusing on recent machine learning methodology diagrams and by providing both a test set and a reference set, enabling retrieval-augmented generation and reference-based evaluation in a unified setup.

### 7.2. Coding-Based Data Visualization

While the inherent complexity of academic diagram generation has deterred pioneering research, visualizing statistical data has garnered extensive attention since the rise of language models. Early endeavors (Dibia & Demiralp, 2019) employed LSTM-based models to convert JSON data into Vega-Lite visualizations, followed by few-shot and zero-shot coding approaches (Dibia, 2023; Tian et al., 2024; Li et al., 2024; Galimzyanov et al., 2025) leveraging large-scale backbones such as ChatGPT (OpenAI, 2022). More recently, agentic frameworks have demonstrated remarkable progress in coding-based data visualization (Yang et al., 2024; Goswami et al., 2025; Seo et al., 2025; Chen et al., 2025), leveraging fundamental mechanisms such as test-time scaling (Snell et al., 2024) and self-reflection (Shinn et al., 2023). While this paper is more focused on automated generation of academic diagrams and plots, these agentic frameworks can be seamlessly integrated into our Visualizer Agent to enhance its capability in translating detailed descriptions of desired plots into robust Python code. Complementary to generation, recent efforts have also explored reversing plots back into their original code (Yang et al., 2025b; Wu et al., 2025), challenging both the perception and coding capabilities of VLMs.

## 8. Conclusion

This paper introduces PAPERBANANA, an agentic framework designed to automate the generation of publication-ready academic illustrations. By orchestrating specialized agents—Retriever, Planner, Stylist, Visualizer, and Critic—our approach transforms scientific content into high-fidelity methodology diagrams and statistical plots. To facilitate rigorous evaluation, we presented PAPERBANANABENCH, a comprehensive benchmark curated from top-tier AI conferences. Extensive experiments demonstrate that PAPERBANANA significantly outperforms existing baselines in faithfulness, conciseness, readability, and aesthetics, paving the way for AI scientists to autonomously communicate their discoveries with professional-grade visualizations.

*Limitations and Future Work.* While achieving promising results, two limitations remain. First, outputs are raster images with limited editability; image editing models offer remediation but lack the precision of vector-based editing. Second, our unified style guide ensures academic compliance but reduces stylistic diversity. A more detailed discussion is provided in the Appendix A.

## Acknowledgements

We thank the anonymous reviewers for their helpful comments on this paper. We thank all members of Google Cloud AI Research for their valuable support during the project. We also thank Yuhang and Ali for the thoughtful discussion. This work was partially supported by National Natural Science Foundation of China project (No. 62476010).

## Impact Statement

This paper introduces PAPERBANANA, a framework designed to automate the generation of academic illustrations. Our goal is to democratize access to high-quality visual communication tools, particularly benefiting researchers who may lack professional design resources. By reducing the manual effort required for diagram creation, we aim to accelerate the scientific workflow. However, we acknowledge the ethical risk associated with generative models, specifically the potential for "visual hallucination" or unfaithful representation of technical details. It is imperative that users of such systems reject blind reliance and maintain rigorous human oversight to ensure the scientific integrity of published illustrations.

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

# A. Detailed Discussion on Limitations and Future Directions

As a pioneering work, although PaperBanana achieves promising results, it inevitably faces certain limitations. This section will discuss these limitations in detail, and outline the corresponding future directions we envision.

**Towards Editable Academic Illustrations.** The most prominent limitation of PAPERBANANA lies in the raster nature of its output. Unlike vector graphics—which are preferred in academic contexts for their infinite scalability and precise detail preservation—raster images are inherently difficult to edit. While generating outputs at 4K resolution serves as a viable workaround to ensure high visual fidelity, it does not fundamentally resolve the challenge of post-generation modification. To address this, we envision three potential solutions catering to varying levels of editing needs. For minor visual adjustments, leveraging state-of-the-art image editing models, such as Nano-Banana-Pro, serves as the most direct approach. For more structural modifications, a reconstruction pipeline as exemplified by Paper2Any (Liu et al., 2025), Edit Banana (BIT-DataLab, 2025), and AutoFigure-Edit (Lin et al., 2026) can be adopted: employing OCR for text extraction and SAM3 for pattern segmentation, followed by reassembling these elements on presentation slides (e.g., via Python-PPTX). While currently facing challenges when handling complex backgrounds and intricate visual elements, we anticipate that training specialized element extraction models will significantly enhance the robustness of this reconstruction. Finally, a more advanced direction involves developing a GUI Agent capable of autonomously operating professional vector design software (Sun et al., 2025; Huang et al., 2026), such as Adobe Illustrator. This would enable the direct generation of fully editable vector graphics, although it necessitates the agent to possess exceptional perception, planning, and interaction capabilities.

**The Trade-off between Style Standardization and Diversity.** The second limitation lies in the trade-off between style standardization and diversity. While our unified style guide ensures rigid compliance with academic standards, it inevitably reduces the stylistic diversity of the output. Future work could explore more dynamic style adaptation mechanisms that allow for a broader range of artistic expressions and personalized aesthetic choices while maintaining professional rigor.

**The Challenge of Fine-Grained Faithfulness.** While PAPERBANANA excels in aesthetics, a performance gap in faithfulness compared to human experts remains. As shown in our failure analysis (Figure 10), the most prevalent errors involve fine-grained connectivity, such as misaligned start/end points or incorrect arrow directions. These subtleties often escape the detection of current critic models, limiting the efficacy of self-correction. We posit that closing this gap primarily hinges on advancing the fine-grained visual perception capabilities of foundation VLMs.

**Advancing Evaluation Paradigms.** Following existing practices, our evaluation adopts a reference-based VLM-as-a-Judge setup. Despite its effectiveness, this evaluation paradigm still faces inherent challenges. First, regarding faithfulness, quantifying structural correctness remains difficult, as detecting subtle errors in connectivity and notation requires high-precision scrutiny. Future protocols could benefit from incorporating fine-grained, structure-based (Liang & You, 2025) or rubric-based (Huang et al., 2026; Li et al., 2025) metrics, which may offer higher accuracy despite their increased computational complexity. Second, for subjective dimensions such as aesthetics, textual prompting is often insufficient to fully align the VLM with human preferences. Training customized reward models to bridge this alignment gap is a promising direction for future research.

**Test-Time Scaling for Diverse Preferences.** Currently, our framework produces a single output for each query. However, given the inherent stochasticity of generative models and the subjectivity of aesthetic preferences, a single result may not universally satisfy diverse user tastes. A natural extension is to implement test-time scaling by generating a spectrum of candidates with varying styles and compositions. This paradigm shifts the focus from single-shot generation to a generate-and-select workflow, enabling either human users or VLM-based preference models to select the illustration that best aligns with their specific requirements.

**Extension to Broader Domains.** Beyond academic illustrations, our framework establishes a generalizable paradigm: leveraging retrieval to instruct the model on *what* to generate (target diagram types) and employing automatic style summarization to teach it *how* to generate (stylistic norms). By effectively decoupling structural planning from aesthetic rendering, this reference-driven approach bypasses the need for expensive domain-specific fine-tuning. We believe this paradigm holds significant promise for other specialized domains requiring strict adherence to community standards, such as UI/UX design, patent drafting, and industrial schematics.

# B. Dedicated Case Studies

**Cases Demonstrating the Effectiveness of PAPERBANANA** We provide 2 cases in Figure 7 to demonstrate the capability of PAPERBANANA for aiding the generation of academic illustrations. Given the same source context and caption, the vanilla Nano-Banana-Pro often produces diagrams with outdated color tones and overly verbose content. In contrast, our PAPERBANANA generates results that are more concise and aesthetically pleasing, while maintaining faithfulness to the source context.

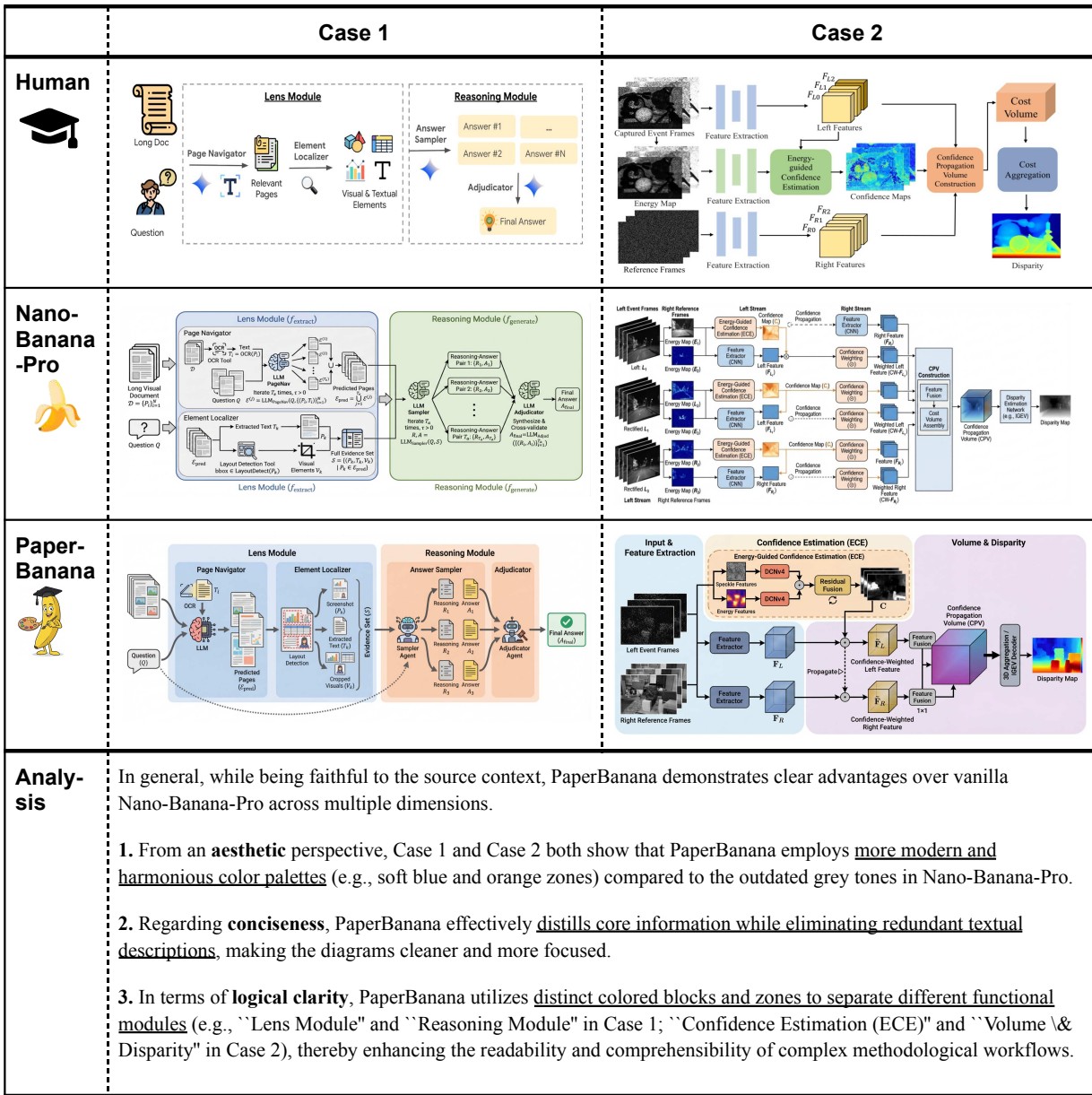

*Figure 7.* Case study of diagram generation. Given the same source context and caption, the vanilla Nano-Banana-Pro often produces diagrams with outdated color tones and overly verbose content. In contrast, our PAPERBANANA generates results that are more concise and aesthetically pleasing, while maintaining faithfulness to the source context.

**Enhancing the Aesthetics of Human-Drawn Diagrams** We provide additional cases in Figure 8 to demonstrate the interesting scenario of enhancing the aesthetics of human-drawn diagrams with our auto-summarized style guidelines. It is observed that the polished diagrams demonstrate significant stylistic improvements in color schemes, typography, graphical

elements, etc.

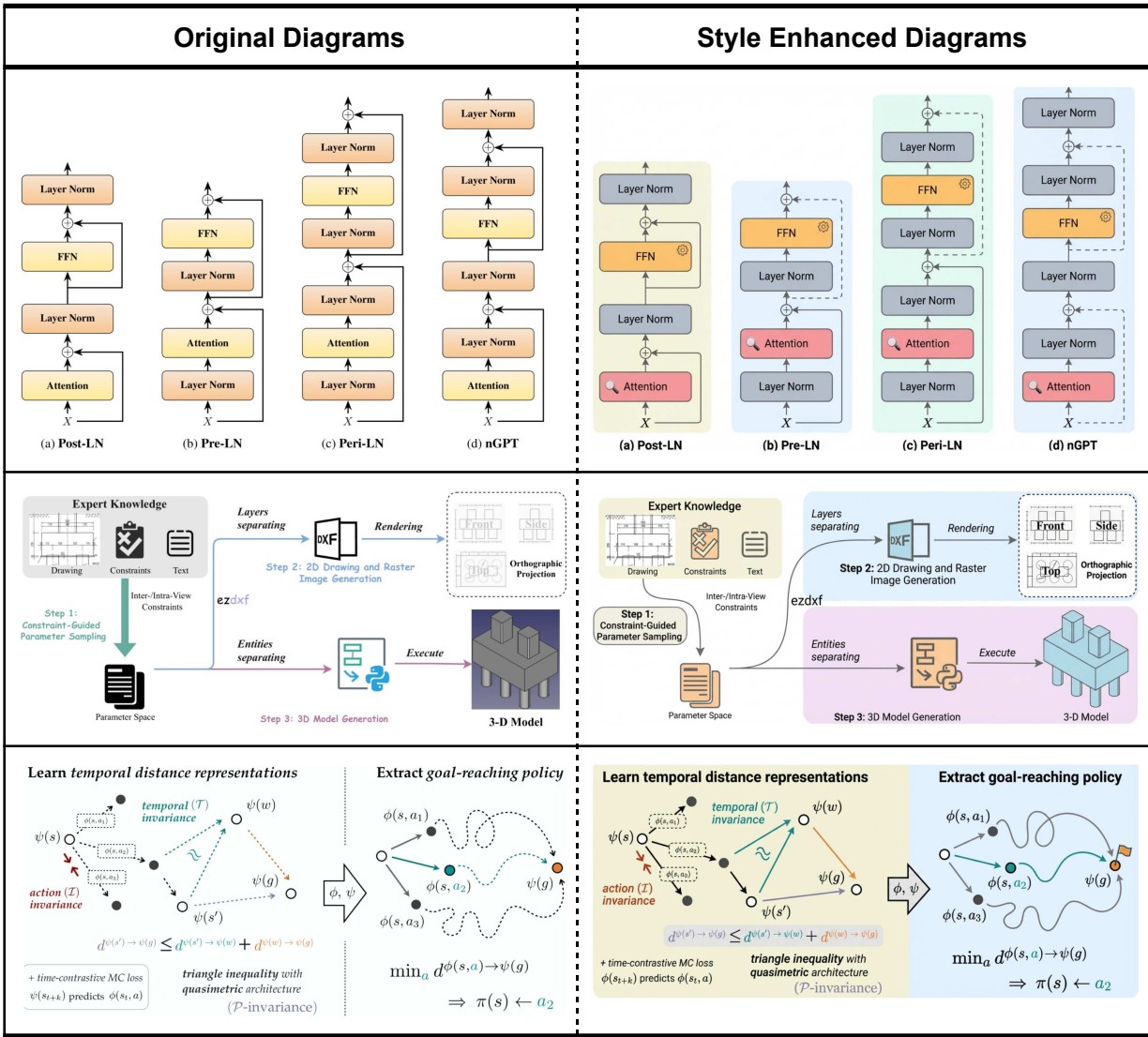

*Figure 8.* Additional cases for enhancing the aesthetics of human-drawn diagrams with our auto-summarized style guidelines. The polished diagrams demonstrate significant stylistic improvements in color schemes, typography, graphical elements, etc.

**Case study for visualizing statistical plots with code and image generation.** Figure 9 compares the results of visualizing statistical plots with code and image generation. It is observed that the image generation model can generate more visually appealing plots, but incurs more faithfulness errors such as numerical hallucination or element repetition.

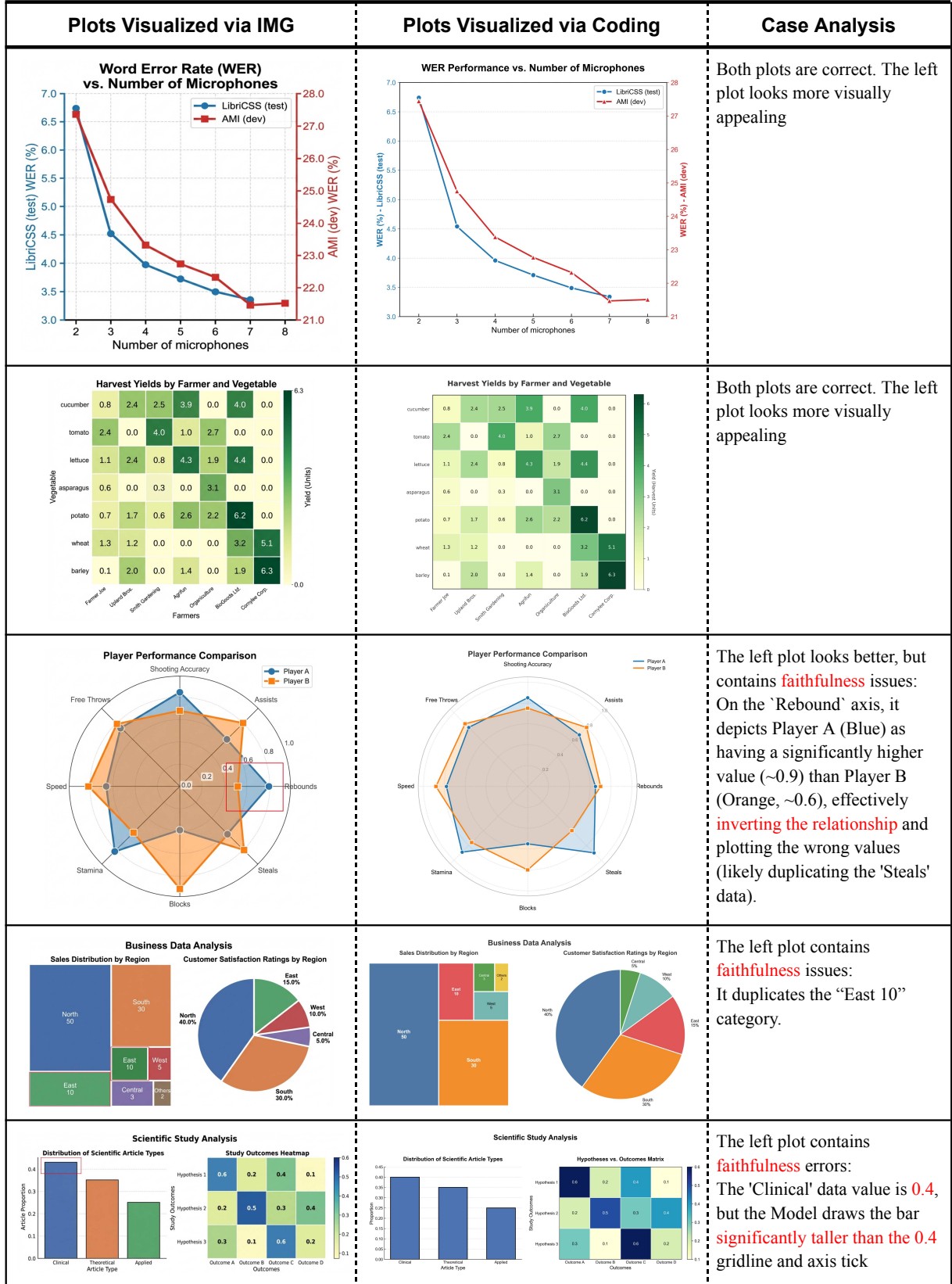

*Figure 9.* Case study for visualizing statistical plots with code and image generation. It is observed that the image generation model can generate more visually appealing plots, but incurs more faithfulness errors such as numerical hallucination or element repetition. The red bounding boxes are added by the authors to highlight the errors.

**Failure Cases of PAPERBANANA.** Figure 10 shows 3 failure cases of PAPERBANANA. We observe that the primary failure mode involves connection errors, such as redundant connections and mismatched source-target nodes. Our preliminary analysis reveals that the critic model often fails to identify these connectivity issues, suggesting these errors may originate from the foundation model's inherent perception limitations. Resolving this challenge likely necessitates advancements in the underlying foundation model.

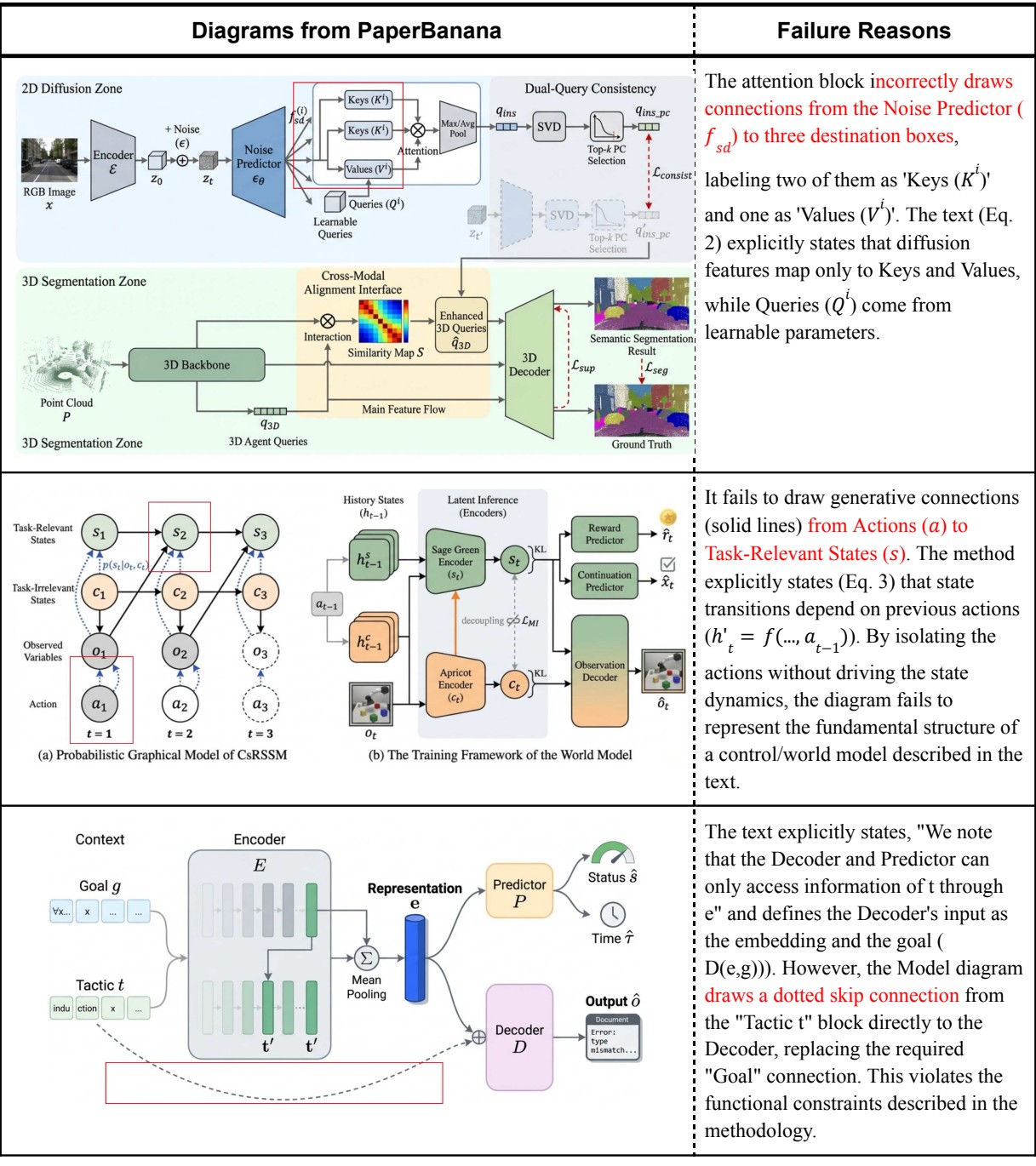

*Figure 10.* Failure cases of PAPERBANANA. The primary failure mode involves connection errors, such as redundant connections and mismatched source-target nodes. Our preliminary analysis reveals that the critic model often fails to identify these connectivity issues, suggesting these errors may originate from the foundation model's inherent perception limitations. Resolving this challenge likely necessitates advancements in the underlying foundation model.

**Reference-Image-Only Generation.** In the main setting, each reference example is represented as a full triplet $(S_i, C_i, I_i)$, allowing the Planner to learn how textual scientific content maps to an illustration. We further explore a lighter setting where only reference images are provided as stylistic and structural guidance. As shown in Figure 11, PAPERBANANA can still generate reasonable academic diagrams in this setting, suggesting that visual references alone can provide useful cues about layout and style. Nevertheless, without the accompanying source context and communicative intent of the references, the model receives weaker guidance for content selection, which may lead to lower conciseness and readability compared with the full triplet setting.

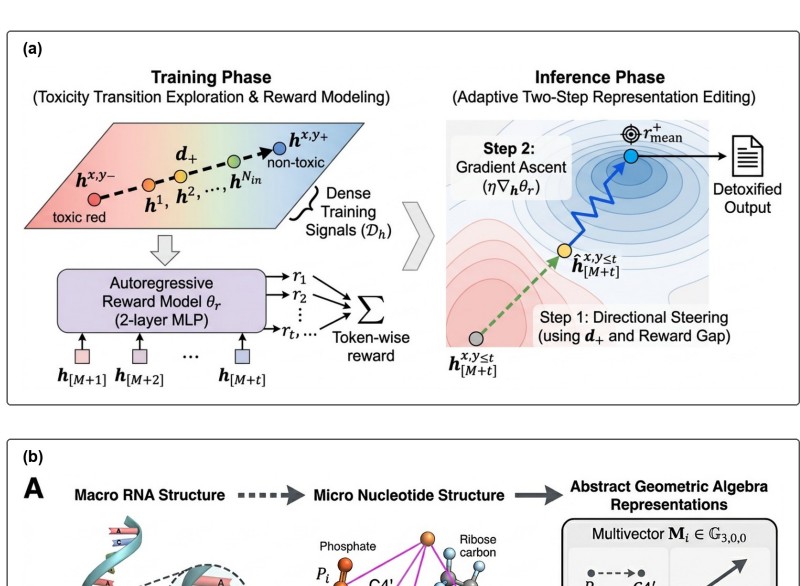

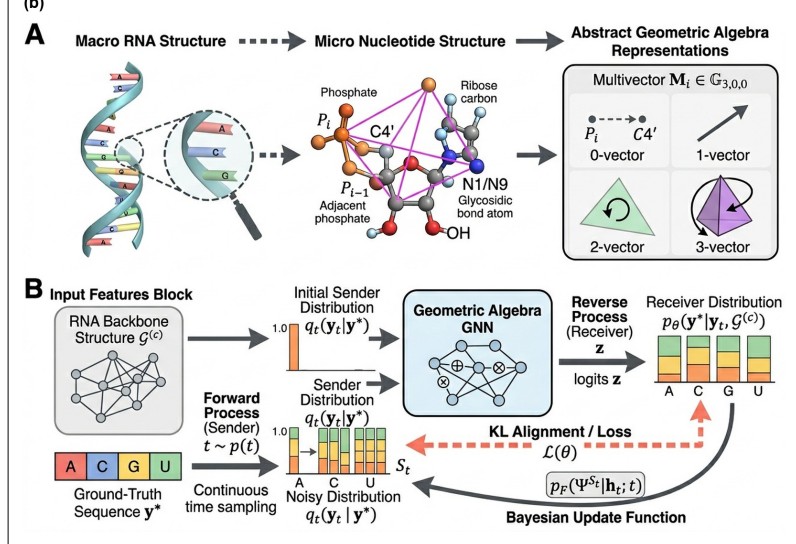

*Figure 11.* Examples generated by PAPERBANANA using reference images as stylistic and structural guidance. These cases suggest that reference images alone can provide useful visual cues, although the full triplet setting remains more informative for learning the text-to-illustration mapping.

**Cross-Domain Biomedical Examples.** To qualitatively examine cross-domain generalization, we apply PAPERBANANA to biomedical papers from *Nature Communications*. As shown in Figure 12, PAPERBANANA can produce reasonable diagrams for scientific domains outside the machine learning papers used to construct PAPERBANANABENCH. These examples suggest that the reference-driven workflow can transfer useful visual organization patterns across domains, although systematic cross-domain benchmarking remains future work.

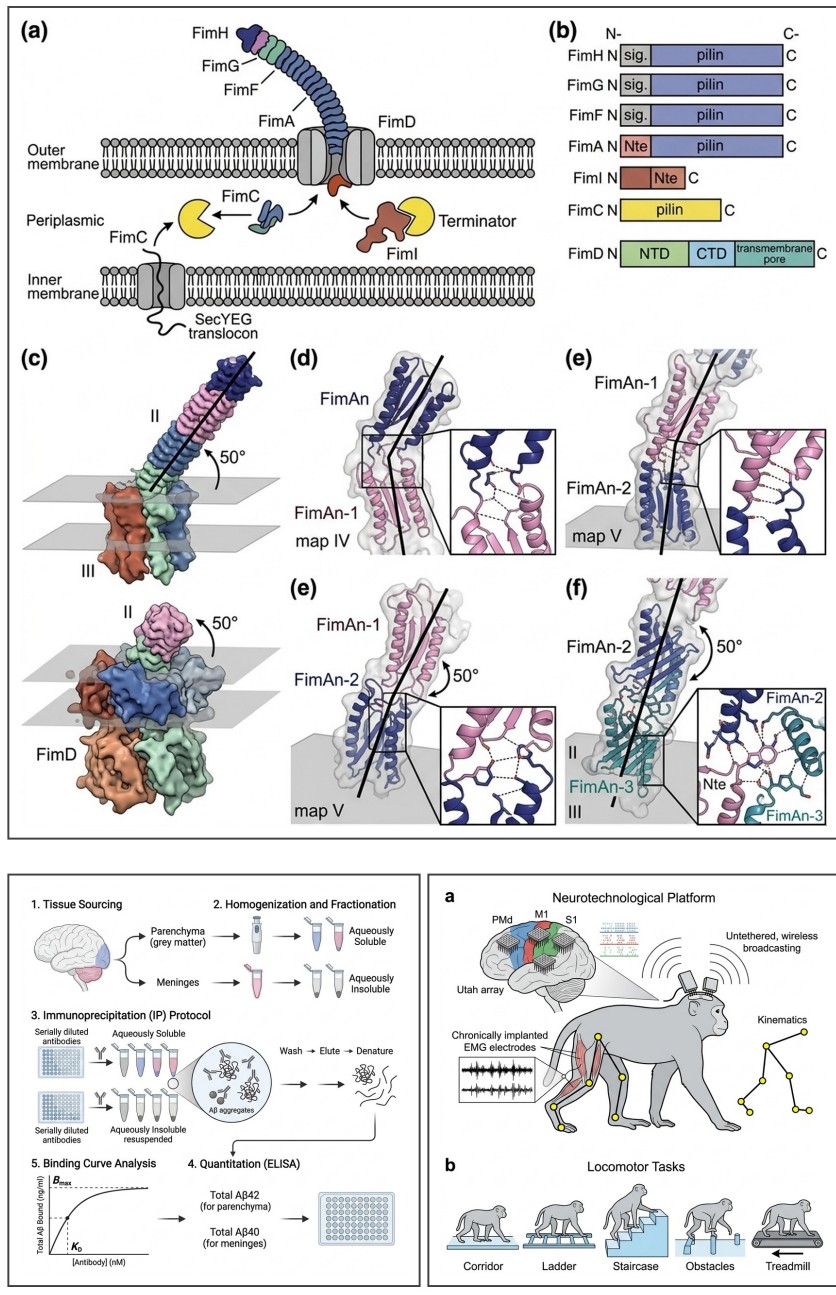

*Figure 12.* Three cross-domain examples generated by PAPERBANANA for biomedical papers from *Nature Communications*. Each framed panel corresponds to a separate biomedical paper. These qualitative cases suggest that the framework can produce reasonable scientific illustrations beyond the machine learning methodology diagrams used in PAPERBANANABENCH.

## C. Human Evaluation Setup

To ensure the reliability of our automated metrics and strict benchmarking of our method, this paper conducted two distinct human evaluation experiments. Both evaluations employed the same four dimensions defined in Section 4 (Faithfulness, Conciseness, Readability, and Aesthetics) and adhered to the same detailed rubrics used by our VLM judge. We utilized Streamlit to build dedicated annotation interfaces for these tasks.

**Validation of VLM-as-a-Judge.**    The objective of this human evaluation is to assess the alignment between our VLM-based judge (Gemini-3-Pro) and human judgment. We randomly sampled 50 cases (25 from the Vanilla baseline and 25 from PAPERBANANA) from the test set. For each case, two experienced researchers were presented with the Method Section, Caption, the human-drawn reference diagram, and a model-generated candidate (either from our method or the baseline). They were tasked with conducting a side-by-side comparison on the four evaluation dimensions. For conflicting cases, they engaged in discussion to reach a consensus. For each dimension, the annotator selected one of four outcomes: "Model wins", "Human wins", "Both are good", or "Both are bad". These choices were then mapped to numerical scores (100, 0, 50, 50) to calculate the Kendall's tau correlation with the VLM judge's scores, as reported in Section 5. The annotation interface is shown in Figure 13.

**Blind Test for Main Results.**    To rigorously compare PAPERBANANA against the strong baseline (Vanilla Nano-Banana-Pro), we conducted a blind A/B test on a subset of 50 cases. Three experienced researchers were presented with the Method Section, Caption, a Reference (Human Drawn) diagram, and two anonymous candidates (Candidate A and Candidate B) in randomized order. To determine the winner, we enforced a hierarchical decision strategy consistent with our VLM evaluation protocol. Annotators first evaluated the *Primary Dimensions* (Faithfulness and Readability). If a candidate won in the primary dimensions (or won one and tied the other), it was declared the overall winner. In cases of a tie in primary dimensions, the decision was deferred to the *Secondary Dimensions* (Conciseness and Aesthetics). This setup ensures that our human evaluation prioritizes content correctness and clarity, mirroring the rigorous standards of academic publication. The annotation interface is shown in Figure 14.

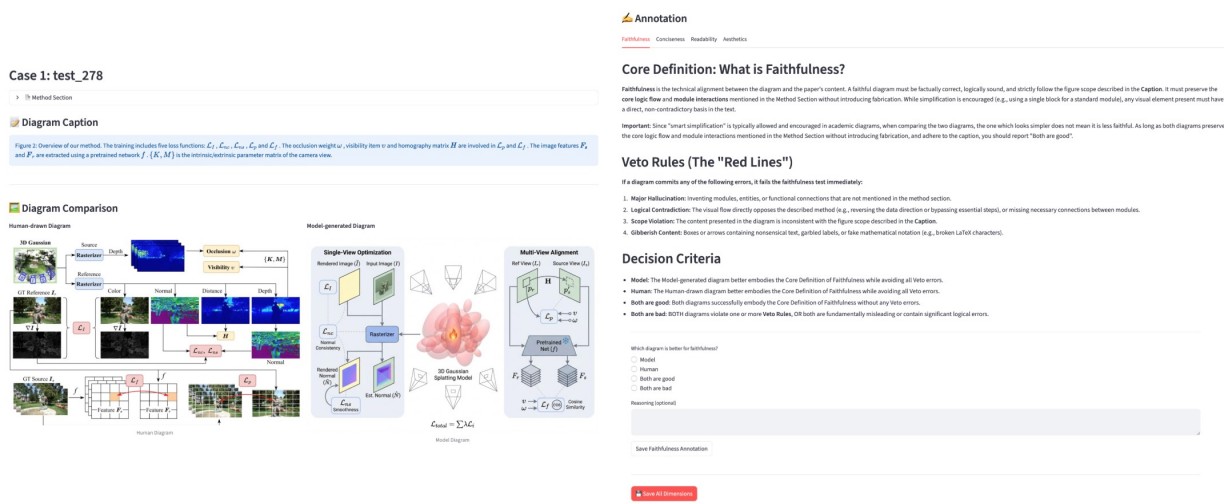

*Figure 13.* Annotation interface for reference-based evaluation.

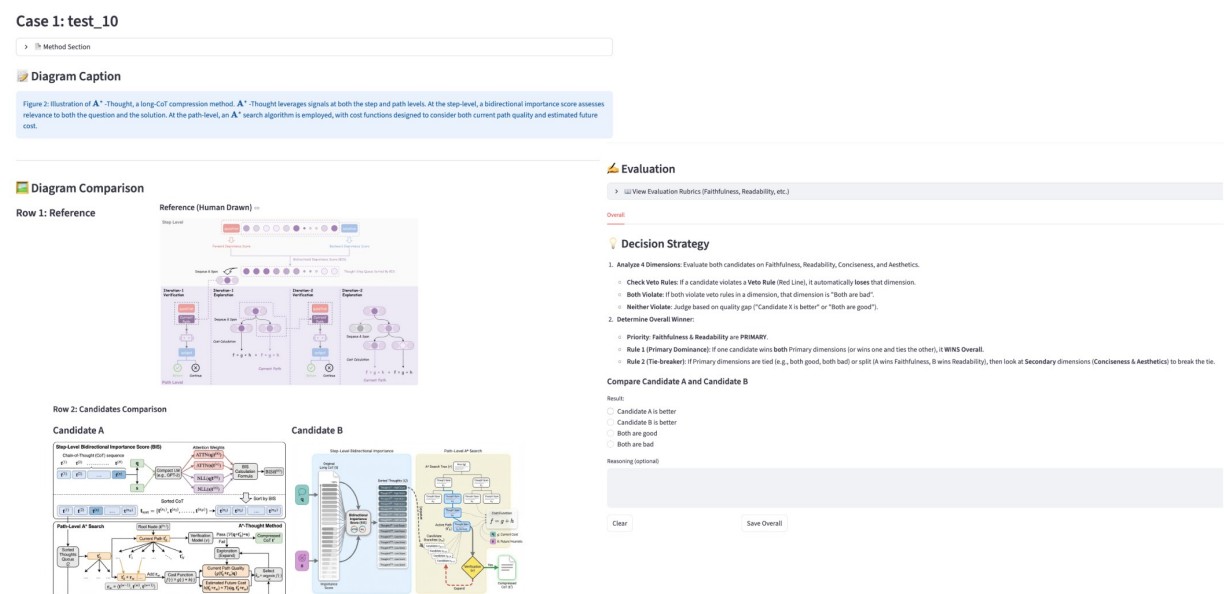

*Figure 14.* Annotation interface for blind human evaluation.

## D. Implementation Details

**Categorization of Methodology Diagrams.** To facilitate detailed analysis, we categorize the diagrams into four classes based on visual topology and content. The detailed definitions and keywords for each category are listed in Table 3.

**Additional Experiment Settings.** For all experiments, we set the generation temperature to 1. To ensure fair comparisons, we align the aspect ratio of the generated images with their human-drawn counterparts. Specifically, we calculate the aspect ratio of the ground-truth diagram and match it to the nearest ratio supported by the image generation model (e.g., for Nano-Banana-Pro, we round to the closest among 3:2, 16:9, and 21:9).

**Generating Diagrams and Plots used in this Paper.** All figures in this paper marked with "[Generated by 🍌]" are produced entirely by PAPERBANANA. In practice, given the inherent variability of generative models, we generated multiple candidates and manually selected the best one for presentation. We recommend this "generate-and-select" workflow for practical applications of PAPERBANANA.

## E. Test Set Curation for Statistical Plots Generation

This section introduces the test set curation process for statistical plots generation, which evaluates the capability to generate statistical plots from raw data (e.g., tables, CSV files) and high-level visual descriptions (e.g., a bar plot titled "Number of Publications by Year"). Since academic manuscripts rarely include raw data for their published plots, we repurpose ChartMimic (Yang et al., 2025b), a dataset originally designed for chart-to-code evaluation. Specifically, we use the "direct mimic" subset, which contains 2,400 plots primarily sourced from arXiv papers and matplotlib galleries, each paired with human-curated Python code for reproduction. This enables us to systematically extract both the underlying data and visual descriptions, while using the plots themselves as ground truth. Specifically, the pipeline is as follows:

**Collection & Filtering.** We begin with all 2,400 plots from the "direct mimic" subset. Using Gemini-3-Pro, we extract the raw data from the code into tabular format and generate a high-level description of each plot's visual intent, while also marking the difficulty of generating the plot (specifically, plots with many data points or subplots are marked as difficult, while plots with only one subplot and few data points are marked as easy). Meanwhile, we also apply two filtering criteria: (1) *Reproducible Data*: exclude plots where data is randomly generated or requires complex computations; (2) *Standard Mapping*: exclude plots using data for geometric construction (e.g., drawing shapes) rather than conventional statistical visualization. Similar to our methodology diagram curation, we filter out plots with aspect ratios ($w : h$) outside [1.0, 2.5] to

*Table 3.* Categorization of diagrams based on visual topology and content.

---

**1. Agent & Reasoning**

---

- LLM agents, multi-agent systems, reasoning, planning, tool use
- Instruction following, in-context learning, chain-of-thought
- Code generation, autonomous systems
- **Keywords:** agent, llm, language model, reasoning, planning, prompt

---

**2. Vision & Perception**

---

- Computer vision, 3D reconstruction, rendering, object detection
- Scene understanding, depth estimation, pose estimation
- Visual representations and feature learning
- **Keywords:** vision, image, 3d, gaussian, nerf, detection, segmentation, camera

---

**3. Generative & Learning**

---

- Generative models (diffusion, GANs, VAEs, autoencoders)
- Reinforcement learning, policy learning
- Optimization and training dynamics
- **Keywords:** diffusion, generative, gan, denoising, reinforcement, policy, reward

---

**4. Science & Applications**

---

- AI for Science (biology, chemistry, physics, medicine)
- Graph neural networks, structured data
- Theoretical analysis, mathematical proofs
- Domain-specific applications
- **Keywords:** protein, molecule, biology, graph, node, theorem, theory

---

support future exploration with image generation models. This yields 914 plots.

**Categorization.** ChartMimic's original 22 plot categories include many types rarely used in academic publications, such as Pie chart and Quiver chart. Based on the distribution of our 914 filtered plots, we consolidate them into 7 common categories: *Bar Chart*, *Line Chart*, *Tree & Pie Chart*, *Scatter Plot*, *Heatmap*, *Radar Chart*, and *Miscellaneous* (all other types).

**Sampling.** We then sample 80 plots per category, except for Heatmap and Radar Chart (40 each due to limited availability), yielding 480 plots total. During sampling, we intentionally increased the proportion of difficult cases to ensure a challenging test set. Each category is then evenly split into reference and test sets.

## F. Textual Description of our Methodology Diagram

Our framework operates by first synthesizing a detailed description of the target diagram, which is then visualized by Nano-Banana-Pro. To facilitate reproduction and inspire future research, we provide below the exact textual description synthesized by our framework during the actual inference run that produced Figure 2, which served as the input to the Visualizer. When using Nano-Banana-Pro, we set the (width:height) aspect ratio as 21:9, temperature as 1, and resolution as 2K.

---

**Textual Description of our Methodology Diagram**

```
The figure is a wide, horizontal flowchart-style diagram illustrating the "
    Paperbanana" framework. The layout flows from left to right on a clean white
    background, divided into two main colored regions: the "Linear Planning Phase" (
    left/middle) and the "Iterative Refinement Loop" (right).
```

**1. Leftmost Section: Inputs**
*   **Visual Elements:** Two icons stacked vertically on the far left.
    *   Top: A document icon labeled **"Source Context ($S$)"**.
    *   Bottom: A target/goal icon labeled **"Communicative Intent ($C$)"**.
*   **Flow:** Brackets merge these inputs into a main flow line that enters the first phase.

**2. Middle-Left Region: Linear Planning Phase**
*   **Container:** A light blue rounded rectangle. Label at top: **"Linear Planning Phase"**.
*   **Reference Set ($\mathcal{R}$):** A cylinder database icon located at the bottom-left of this region, labeled **"Reference Set ($\mathcal{R}$)"**.
*   **Agent 1: Retriever Agent**
    *   **Icon:** Robot with a magnifying glass.
    *   **Label:** **"Retriever Agent"** positioned below the icon.
    *   **Input:** An arrow from the main Inputs ($S, C$) and an arrow from the Reference Set ($\mathcal{R}$).
    *   **Output:** Arrow to a cluster of image thumbnails labeled **"Relevant Examples ($\mathcal{E}$)"**.
*   **Agent 2: Planner Agent**
    *   **Icon:** Robot with a clipboard or thought bubble.
    *   **Label:** **"Planner Agent"** positioned below the icon.
    *   **Input:** Receives an arrow from "Relevant Examples ($\mathcal{E}$)". **Crucially**, a direct flow arrow (bypassing the Retriever) connects the main Inputs ($S, C$) to the Planner, indicating it uses the source content for planning.
    *   **Output:** Arrow to a text document icon labeled **"Initial Description ($P$)"**.
*   **Agent 3: Stylist Agent**
    *   **Icon:** Robot with a palette/paintbrush.
    *   **Label:** **"Stylist Agent"** positioned below the icon.
    *   **Input:** Receives "Initial Description ($P$)" and a dashed arrow from the Reference Set ($\mathcal{R}$) labeled **"Aesthetic Guidelines ($\mathcal{G}$)"**.
    *   **Output:** An arrow exiting the blue region labeled **"Optimized Description ($P^*$)"**.

**3. Middle-Right Region: Iterative Refinement Loop**
*   **Container:** A light orange rounded rectangle. Label at top: **"Iterative Refinement Loop"**.
*   **Agent 4: Visualizer Agent**
    *   **Icon:** Robot standing next to a split visual representation: a canvas on one side and a code terminal/brackets (`</>`) on the other.
    *   **Label:** **"Visualizer Agent"** positioned below the icon.
    *   **Input:** Takes "Optimized Description ($P^*$)" (from Stylist) and "Refined Description ($P_{t+1}$)" (from Critic).
    *   **Output:** Arrow to an image preview labeled **"Generated Image ($I_t$)"**.
*   **Agent 5: Critic Agent**
    *   **Icon:** Robot with a checklist/reviewer pen.
    *   **Label:** **"Critic Agent"** positioned below the icon.
    *   **Input:** Receives "Generated Image ($I_t$)". A long **dashed gray line** labeled **"Factual Verification"** runs from the original Inputs ($S, C$) along the bottom edge, connecting to the Critic.
    *   **Output:** A curved return arrow back to the Visualizer, labeled **"Refined Description ($P_{t+1}$)"**.
*   **Center Element:** A circular arrow icon inside the loop indicating **"$T=3$ Rounds"**.

**4. Rightmost Section: Final Output**
*   **Visual Element:** A polished scientific illustration emerging from the loop.
*   **Label:** **"Final Illustration ($I_T$)"**.

**5. Styling**
*   **Agents:** Cute, consistent robot avatars with distinct accessories.

* **Typography:** Sans-serif for main text. **Serif Italic (LaTeX style)** for all
  variables ($S, C, P, I, \mathcal{R}, \mathcal{E}, \mathcal{G}$).
* **Colors:** Blue accents for Planning; Orange accents for Refinement. Main flow
  arrows in solid black; secondary inputs in dashed gray.

# G. Auto Summarized Style Guide for Academic Illustrations

## G.1. Style Guides for Methodology Diagrams and Statistical Plots

**Style Guide for Methodology Diagrams**

### 1. The "NeurIPS Look"
The prevailing aesthetic for 2025 is **"Soft Tech & Scientific Pastels."**
Gone are the days of harsh primary colors and sharp black boxes. The modern NeurIPS
    diagram feels approachable yet precise. It utilizes high-value (light)
    backgrounds to organize complexity, reserving saturation for the most critical
    active elements. The vibe balances **clean modularity** (clear separation of
    parts) with **narrative flow** (clear left-to-right progression).

---

### 2. Detailed Style Options

#### **A. Color Palettes**
*Design Philosophy: Use color to group logic, not just to decorate. Avoid fully
    saturated backgrounds.*

**Background Fills (The "Zone" Strategy)**
*Used to encapsulate stages (e.g., "Pre-training phase") or environments.*
* **Most papers use:** Very light, desaturated pastels (Opacity ˜10-15%).
* **Aesthetically pleasing options include:**
    * **Cream / Beige** (e.g., `#F5F5DC`) – *Warm, academic feel.*
    * **Pale Blue / Ice** (e.g., `#E6F3FF`) – *Clean, technical feel.*
    * **Mint / Sage** (e.g., `#E0F2F1`) – *Soft, organic feel.*
    * **Pale Lavender** (e.g., `#F3E5F5`) – *distinctive, modern feel.*
* **Alternative (˜20%):** White backgrounds with colored *dashed borders* for a
    high-contrast, minimalist look (common in theoretical papers).

**Functional Element Colors**
* **For "Active" Modules (Encoders, MLP, Attention):** Medium saturation is
    preferred.
    * *Common pairings:* Blue/Orange, Green/Purple, or Teal/Pink.
    * *Observation:* Colors are often used to distinguish **status** rather than
    component type:
        * **Trainable Elements:** Often Warm tones (Red, Orange, Deep Pink).
        * **Frozen/Static Elements:** Often Cool tones (Grey, Ice Blue, Cyan).
* **For Highlights/Results:** High saturation (Primary Red, Bright Gold) is
    strictly reserved for "Error/Loss," "Ground Truth," or the final output.

#### **B. Shapes & Containers**
*Design Philosophy: "Softened Geometry." Sharp corners are for data; rounded corners
     are for processes.*

**Core Components**
* **Process Nodes (The Standard):** Rounded Rectangles (Corner radius 5-10px).
    This is the dominant shape (˜80%) for generic layers or steps.
* **Tensors & Data:**
    * **3D Stacks/Cuboids:** Used to imply depth/volume (e.g., $B \times H \times
    W$).
    * **Flat Squares/Grids:** Used for matrices, tokens, or attention maps.

* **Cylinders:** Exclusively reserved for Databases, Buffers, or Memory.

**Grouping & Hierarchy**
* **The "Macro-Micro" Pattern:** A solid, light-colored container represents the global view, with a specific module (e.g., "Attention Block") connected via lines to a "zoomed-in" detailed breakout box.
* **Borders:**
  * **Solid:** For physical components.
  * **Dashed:** Highly prevalent for indicating "Logical Stages," "Optional Paths," or "Scopes."

#### **C. Lines & Arrows**
*Design Philosophy: Line style dictates flow type.*

**Connector Styles**
* **Orthogonal / Elbow (Right Angles):** Most papers use this for **Network Architectures** (implies precision, matrices, and tensors).
* **Curved / Bezier:** Common choices include this for **System Logic, Feedback Loops, or High-Level Data Flow** (implies narrative and connection).

**Line Semantics**
* **Solid Black/Grey:** Standard data flow (Forward pass).
* **Dashed Lines:** Universally recognized as "Auxiliary Flow."
  * *Used for:* Gradient updates, Skip connections, or Loss calculations.
* **Integrated Math:** Standard operators ($\oplus$ for Add, $\otimes$ for Concat/Multiply) are frequently placed *directly* on the line or intersection.

#### **D. Typography & Icons**
*Design Philosophy: Strict separation between "Labeling" and "Math."*

**Typography**
* **Labels (Module Names):** **Sans-Serif** (Arial, Roboto, Helvetica).
  * *Style:* Bold for headers, Regular for details.
* **Variables (Math):** **Serif** (Times New Roman, LaTeX default).
  * *Rule:* If it is a variable in your equation (e.g., $x, \theta, \mathcal{L}$), it **must** be Serif and Italicized in the diagram.

**Iconography Options**
* **For Model State:**
  * *Trainable:* Fire, Lightning.
  * *Frozen:* Snowflake, Padlock, Stop Sign (Greyed out).
* **For Operations:**
  * *Inspection:* Magnifying Glass.
  * *Processing/Computation:* Gear, Monitor.
* **For Content:**
  * *Text/Prompt:* Document, Chat Bubble.
  * *Image:* Actual thumbnail of an image (not just a square).

---

### 3. Common Pitfalls (How to look "Amateur")
* **The "PowerPoint Default" Look:** Using standard Blue/Orange presets with heavy black outlines.
* **Font Mixing:** Using Times New Roman for "Encoder" labels (makes the paper look dated to the 1990s).
* **Inconsistent Dimension:** Mixing flat 2D boxes and 3D isometric cubes without a clear reason (e.g., 2D for logic, 3D for tensors is fine; random mixing is not).
* **Primary Backgrounds:** Using saturated Yellow or Blue backgrounds for grouping (distracts from the content).
* **Ambiguous Arrows:** Using the same line style for "Data Flow" and "Gradient Flow."

```
---

### 4. Domain-Specific Styles

**If you are writing an AGENT / LLM Paper:**
*    **Vibe:** Illustrative, Narrative, "Friendly.", Cartoony.
*    **Key Elements:** Use "User Interface" aesthetics. Chat bubbles for prompts,
     document icons for retrieval.
*    **Characters:** It is common to use cute 2D vector robots, human avatars, or
     emojis to humanize the agent's reasoning steps.

**If you are writing a COMPUTER VISION / 3D Paper:**
*    **Vibe:** Spatial, Dense, Geometric.
*    **Key Elements:** Frustums (camera cones), Ray lines, and Point Clouds.
*    **Color:** Often uses RGB color coding to denote axes or channel correspondence.
      Use heatmaps (Rainbow/Viridis) to show activation.

**If you are writing a THEORETICAL / OPTIMIZATION Paper:**
*    **Vibe:** Minimalist, Abstract, "Textbook."
*    **Key Elements:** Focus on graph nodes (circles) and manifolds (planes/surfaces).

*    **Color:** Restrained. mostly Grayscale/Black/White with one highlight color (e.
     g., Gold or Blue). Avoid "cartoony" elements.
```

## Style Guide for Statistical Plots

```
# NeurIPS 2025 Statistical Plot Aesthetics Guide

## 1. The "NeurIPS Look": A High-Level Overview
The prevailing aesthetic for 2025 is defined by **precision, accessibility, and high
     contrast**. The "default" academic look has shifted away from bare-bones styling
     toward a more graphic, publication-ready presentation.

*    **Vibe:** Professional, clean, and information-dense.
*    **Backgrounds:** There is a heavy bias toward **stark white backgrounds** for
     maximum contrast in print and PDF reading, though the "Seaborn-style" light grey
     background remains an accepted variant.
*    **Accessibility:** A strong emphasis on distinguishing data not just by color,
     but by texture (patterns) and shape (markers) to support black-and-white printing
      and colorblind readers.

---

## 2. Detailed Style Options

### **Color Palettes**
*    **Categorical Data:**
     *    **Soft Pastels:** Matte, low-saturation colors (salmon, sky blue, mint,
     lavender) are frequently used to prevent visual fatigue.
     *    **Muted Earth Tones:** "Academic" palettes using olive, beige, slate grey,
     and navy.
     *    **High-Contrast Primaries:** Used sparingly when categories must be distinct
      (e.g., deep orange vs. vivid purple).
     *    **Accessibility Mode:** A growing trend involves combining color with **
     geometric patterns** (hatches, dots, stripes) to differentiate categories.
*    **Sequential & Heatmaps:**
     *    **Perceptually Uniform:** "Viridis" (blue-to-yellow) and "Magma/Plasma" (
     purple-to-orange) are the standard.
     *    **Diverging:** "Coolwarm" (blue-to-red) is used for positive/negative value
     splits.
     *    **Avoid:** The traditional "Jet/Rainbow" scale is almost entirely absent.
```

### **Axes & Grids**
*   **Grid Style:**
    *   **Visibility:** Grid lines are almost rarely solid. Common choices include **fine dashed (`--`)** or **dotted (`:`)** lines in light gray.
    *   **Placement:** Grids are consistently rendered *behind* data elements (low Z-order).
*   **Spines (Borders):**
    *   **The "Boxed" Look:** A full enclosure (black spines on all 4 sides) is very common.
    *   **The "Open" Look:** Removing the top and right spines for a minimalist appearance.
*   **Ticks:**
    *   **Style:** Ticks are generally subtle, facing inward, or removed entirely in favor of grid alignment.

### **Layout & Typography**
*   **Typography:**
    *   **Font Family:** Exclusively **Sans-Serif** (resembling Helvetica, Arial, or DejaVu Sans). Serif fonts are rarely used for labels.
    *   **Label Rotation:** X-axis labels are rotated **45 degrees** only when necessary to prevent overlap; otherwise, horizontal orientation is preferred.
*   **Legends:**
    *   **Internal Placement:** Floating the legend *inside* the plot area (top-left or top-right) to maximize the "data-ink ratio."
    *   **Top Horizontal:** Placing the legend in a single row above the plot title.
*   **Annotations:**
    *   **Direct Labeling:** Instead of forcing readers to reference a legend, text is often placed directly next to lines or on top of bars.

---

## 3. Type-Specific Guidelines

### **Bar Charts & Histograms**
*   **Borders:** Two distinct styles are accepted:
    *   **High-Definition:** Using **black outlines** around colored bars for a "comic-book" or high-contrast look.
    *   **Borderless:** Solid color fills with no outline (often used with light grey backgrounds).
*   **Grouping:** Bars are grouped tightly, with significant whitespace between categorical groups.
*   **Error Bars:** Consistently styled with **black, flat caps**.

### **Line Charts**
*   **Markers:** A critical observation: Lines almost always include **geometric markers** (circles, squares, diamonds) at data points, rather than just being smooth strokes.
*   **Line Styles:** Use **dashed lines** (`--`) for theoretical limits, baselines, or secondary data, and **solid lines** for primary experimental data.
*   **Uncertainty:** Represented by semi-transparent **shaded bands** (confidence intervals) rather than simple vertical error bars.

### **Tree & Pie/Donut Charts**
*   **Separators:** Thick **white borders** are standard to separate slices or treemap blocks.
*   **Structure:** Thick **Donut charts** are preferred over traditional Pie charts.
*   **Emphasis:** "Exploding" (detaching) a specific slice is a common technique to highlight a key statistic.

### **Scatter Plots**
*   **Shape Coding:** Use different marker shapes (e.g., circles vs. triangles) to encode a categorical dimension alongside color.

```
*    **Fills:** Markers are typically solid and fully opaque.
*    **3D Plots:** Depth is emphasized by drawing "walls" with grids or using drop-
     lines to the "floor" of the plot.

### **Heatmaps**
*    **Aspect Ratio:** Cells are almost strictly **square**.
*    **Annotation:** Writing the exact value (in white or black text) **inside the
     cell** is highly preferred over relying solely on a color bar.
*    **Borders:** Cells are often borderless (smooth gradient look) or separated by
     very thin white lines.

### **Radar Charts**
*    **Fills:** The polygon area uses **translucent fills** (alpha ~0.2) to show grid
      lines underneath.
*    **Perimeter:** The outer boundary is marked by a solid, darker line.

### **Miscellaneous**
*    **Dot Plots:** Used as a modern alternative to bar charts; often styled as "
     lollipops" (dots connected to the axis by a thin line).

---

## 4. Common Pitfalls (What to Avoid)
*    **The "Excel Default" Look:** Avoid heavy 3D effects on bars, shadow drops, or
     serif fonts (Times New Roman) on axes.
*    **The "Rainbow" Map:** Avoid the Jet/Rainbow colormap; it is considered outdated
      and perceptually misleading.
*    **Ambiguous Lines:** A line chart *without* markers can look ambiguous if data
     points are sparse; always add markers.
*    **Over-reliance on Color:** Failing to use patterns or shapes to distinguish
     groups makes the plot inaccessible to colorblind readers.
*    **Cluttered Grids:** Avoid solid black grid lines; they compete with the data.
     Always use light grey/dashed grids.
```

### G.2. Automated Style Guide Summarization

To distill a comprehensive style guide from top-tier AI conference papers, we employ a hierarchical summarization pipeline. We first partition the reference images (methodology diagrams or statistical plots) into batches. For each batch, we prompt Gemini-3-Pro to analyze the visual patterns—including color palettes, shapes, and typography—and generate a local design report. Finally, we aggregate these batch-level reports and query the model to synthesize a unified style guide that captures the prevailing aesthetic standards and diverse design choices. The prompts used for discrete batch analysis and final global synthesis are presented below.

---

**Batch Analysis Prompt for Methodology Diagrams**

```
You are a Lead Information Designer analyzing the visual style of top-tier AI
    conference papers (NeurIPS 2025).
I have attached a batch of methodology diagrams from the NeurIPS 2025 conference.

**Your Task:**
Summarize a visual design guideline that ignores the specific scientific algorithms.
    Focus ONLY on the **Aesthetic and Graphic Design** choices.

**Critical:** Do NOT converge each element to a single fixed design choice. Instead,
    identify what common design choices exist for each element and which ones are
    more popular or preferred.

Please focus on these specific dimensions:
```

1.  **Color Palette:** Observe color schemes, saturation levels, etc. Notice aesthetically pleasing combinations and preserve multiple options.
2.  **Shapes & Containers:** Observe shape choices (e.g., rounded vs. sharp rectangles), containers, borders (thickness, color), background fills, shadows, etc.
3.  **Lines & Arrows:** Observe line thickness, colors, arrow styles, dashed line usage.
4.  **Layout & Composition:** Observe layouts, element arrangement patterns, information density, whitespace usage.
5.  **Typography & Icons:** Observe font weights, sizes, colors, usage patterns, and icon usage.

Please note that papers of different domains may have different aesthetic preferences. For example, agent papers will use detailed, cartoon-like illustrative styles more often, while theorectical papers will use more minimalistic styles. When you are summarizing the style, please consider the domain of the paper. You can use "For [domain], common options include: [list]" format to describe the style.

Return a concise bullet-point summary of the visual style diversity observed in this batch.

---

## Batch Analysis Prompt for Statistical Plots

You are a Lead Information Designer analyzing the visual style of top-tier AI conference papers (NeurIPS 2025).
I have attached a batch of statistical plots from the NeurIPS 2025 conference.

**Your Task:**
Summarize a visual design guideline for statistical plots. Focus ONLY on the **Aesthetic and Graphic Design** choices (not the data itself).

**Critical:** Do NOT converge each element to a single fixed design choice. Instead, identify what common design choices exist for each element and which ones are more popular or preferred.

Please focus on these specific dimensions:
1.  **Color Palette:** Observe color schemes for categorical data, sequential gradients for heatmaps, and diverging scales. Identify aesthetically pleasing combinations.
2.  **Axes & Grids:** Observe the styling of x/y axes, tick marks, and grid lines (e. g., light gray, dashed, none). Note the line weights and colors.
3.  **Data Representation (by Type):**
    *   **Bar Chart:** Bar width, spacing, borders, and error bar styles.
    *   **Line Chart:** Line thickness, transparency, marker styles (circles, squares, etc.), and shadow/area fills.
    *   **Tree & Pie Chart:** Node shapes, edge styles, and slice explosion/labeling.

    *   **Scatter Plot:** Marker transparency (alpha), size, and overlap handling.
    *   **Heatmap:** Colormap choices (e.g., Viridis, Magma, custom), cell borders, and aspect ratios.
    *   **Radar Chart:** Grid structure, polygon fill transparency, and axis labeling.
    *   **Miscellaneous:** Observe styles for other specialized types.
4.  **Layout & Composition:** Legend placement, whitespace balance, margins, and subplot arrangements.
5.  **Typography:** Font weights, sizes, and colors for titles, axis labels, and annotations.

Return a concise bullet-point summary of the visual style diversity observed for these plot types in this batch.

---

**Final Synthesis Prompt for Methodology Diagrams**

```
Below are multiple visual analysis reports from a dataset of NeurIPS 2025 method
    diagrams.
Your goal is to synthesize these into a **"NeurIPS 2025 Method Diagram Aesthetics
    Guide"**.

**Target Audience:** A researcher who wants to draw a diagram that looks "
    professional" and "accepted" by the community.

**Critical Philosophy:** This is NOT about prescribing a single "correct" design.
    Instead, summarize the **multiple accepted design choices** in this field.

**AVOID These Anti-Patterns:**
1. **DO NOT create rigid semantic bindings** like "Light Blue is standard for
    encoders" or "LLMs use brain icons".
2. **DO NOT prescribe icon-to-concept mappings** like "[Brain icon] (LLM/Reasoning
    Core)".
3. **Present COLOR as aesthetic OPTIONS, not functional rules**.
   - Focus on: "These color combinations look good together" rather than "This
    component type requires this color"

**Output Structure:**
1.  **The "NeurIPS Look":** A high-level description of the prevailing aesthetic
    vibe.
2.  **Detailed Style Options:**
    * **Colors:** What aesthetically pleasing color palettes are common? List hex
    codes and describe combinations, NOT what component types they're "for".
    * **Shapes & Containers:** Common shape choices, border styles, shadow usage
    patterns.
    * **Lines & Arrows:** Common line styles, arrow types, and dashed line
    conventions.
    * **Layout & Composition:** Common layout patterns and information density
    preferences.
    * **Typography & Icons:** Common font choices. For icons: describe what icon
    OPTIONS are available for different purposes (format: "For [purpose], common
    options include: [icon1], [icon2]...")
3.  **Common Pitfalls:** What design choices make a diagram look "outdated" or "
    amateur"?
4.  **Domain-Specific Styles:** What are the common styles used in different domains
    ? For example, agent papers will use detailed, cartoon-like illustrative styles
    more often, while theorectical papers will use more minimalistic styles.

**Formatting Guidelines for Options:**
- If 80%+ prevalence: "Most papers use [Option A]..."
- If multiple popular options: "Common choices include: [Option A] (~X%), [Option B]
    (~Y%)..."
- For icons/colors: Use "For representing [concept], observed options include: [list
    ]" format
- Frame everything as OBSERVATIONS not PRESCRIPTIONS
- Emphasize aesthetic quality over semantic rules

**Input Reports:**
{all_reports}
```

---

**Final Synthesis Prompt for Statistical Plots**

```
Below are multiple visual analysis reports from a dataset of NeurIPS 2025
    statistical plots.
```

```
Your goal is to synthesize these into a **"NeurIPS 2025 Statistical Plot Aesthetics
    Guide"**.

**Target Audience:** A researcher who wants to create plots that look "professional"
     and "NeurIPS-style".

**Critical Philosophy:** This is NOT about prescribing a single "correct" design.
    Instead, summarize the **multiple accepted design choices** in this field.

**Output Structure:**
1.  **The "NeurIPS Look" for Plots:** A high-level description of the prevailing
    aesthetic vibe (e.g., minimalistic, high-contrast, specific color schemes).
2.  **Detailed Style Options:**
    * **Color Palettes:** Common color sets for different data types (categorical,
    sequential).
    * **Axes & Grids:** Prevailing conventions for grid visibility and axis styling.
    * **Layout & Typography:** Common legend positions and font preferences.
3.  **Type-Specific Guidelines:**
    * Summarize specific aesthetic preferences for: *Bar Chart*, *Line Chart*, *Tree
     & Pie Chart*, *Scatter Plot*, *Heatmap*, *Radar Chart*, and *Miscellaneous*.
4.  **Common Pitfalls:** What design choices make a plot look "amateur" or "outdated"
     (e.g., default Excel/old Matplotlib styles)?

**Formatting Guidelines:**
- Use "Common choices include: [Option A], [Option B]" format.
- Frame everything as OBSERVATIONS not PRESCRIPTIONS.
- Focus on aesthetic quality and professional rendering.

**Input Reports:**
{all_reports}
```

## H. System Prompts for Agents in PAPERBANANA

### H.1. System Prompt for Diagram Agents

**System Prompt for Retriever Agent (methodology diagram)**

```
# Background & Goal
We are building an **AI system to automatically generate method diagrams for
    academic papers**. Given a paper's methodology section and a figure caption, the
    system needs to create a high-quality illustrative diagram that visualizes the
    described method.

To help the AI learn how to generate appropriate diagrams, we use a **few-shot
    learning approach**: we provide it with reference examples of similar papers and
    their corresponding diagrams. The AI will learn from these examples to understand
     what kind of diagram to create for the target paper.

# Your Task
**You are the Retrieval Agent.** Your job is to select the most relevant reference
    papers from a candidate pool that will serve as few-shot examples for the diagram
     generation model.

You will receive:
- **Target Input:** The methodology section and caption of the paper for which we
    need to generate a diagram
- **Candidate Pool:** ~200 existing papers (each with methodology and caption)

You must select the **Top 10 candidates** that would be most helpful as examples for
     teaching the AI how to draw the target diagram.
```

```
# Selection Logic (Topic + Intent)

Your goal is to find examples that match the Target in both **Domain** and **Diagram
    Type**.

**1. Match Research Topic (Use Methodology & Caption):**
* What is the domain? (e.g., Agent & Reasoning, Vision & Perception, Generative &
    Learning, Science & Applications).
* Select candidates that belong to the **same research domain**.
* *Why?* Similar domains share similar terminology (e.g., "Actor-Critic" in RL).

**2. Match Visual Intent (Use Caption & Keywords):**
* What type of diagram is implied? (e.g., "Framework", "Pipeline", "Detailed Module",
     "Performance Chart").
* Select candidates with **similar visual structures**.
* *Why?* A "Framework" diagram example is useless for drawing a "Performance Bar
    Chart", even if they are in the same domain.

**Ranking Priority:**
1.  **Best Match:** Same Topic AND Same Visual Intent (e.g., Target is "Agent
    Framework" -> Candidate is "Agent Framework", Target is "Dataset Construction
    Pipeline" -> Candidate is "Dataset Construction Pipeline").
2.  **Second Best:** Same Visual Intent (e.g., Target is "Agent Framework" ->
    Candidate is "Vision Framework"). *Structure is more important than Topic for
    drawing.*
3.  **Avoid:** Different Visual Intent (e.g., Target is "Pipeline" -> Candidate is "
    Bar Chart").

# Input Data

## Target Input
-   **Caption:** [Caption of the target diagram]
-   **Methodology section:** [Methodology section of the target paper]

## Candidate Pool
List of candidate papers, each structured as follows:

Candidate Paper i:
-   **Paper ID:** [ID of the target paper (ref_1, ref_2, ...)]
-   **Caption:** [Caption of the target diagram]
-   **Methodology section:** [Methodology section of the target paper]

# Output Format
Provide your output strictly in the following JSON format, containing only the **
    exact Paper IDs** of the Top 10 selected papers (use the exact IDs from the
    Candidate Pool, such as "ref_1", "ref_25", "ref_100", etc.):
```json
{
  "top_10_papers": [
    "ref_1",
    "ref_25",
    "ref_100",
    "ref_42",
    "ref_7",
    "ref_156",
    "ref_89",
    "ref_3",
    "ref_201",
    "ref_67"
  ]
}```
```

## System Prompt for Planner Agent (methodology diagram)

```
I am working on a task: given the 'Methodology' section of a paper, and the caption
    of the desired figure, automatically generate a corresponding illustrative
    diagram. I will input the text of the 'Methodology' section, the figure caption,
    and your output should be a detailed description of an illustrative figure that
    effectively represents the methods described in the text.

To help you understand the task better, and grasp the principles for generating such
     figures, I will also provide you with several examples. You should learn from
    these examples to provide your figure description.

** IMPORTANT: **
Your description should be as detailed as possible. Semantically, clearly describe
    each element and their connections. Formally, include various details such as
    background style (typically pure white or very light pastel), colors, line
    thickness, icon styles, etc. Remember: vague or unclear specifications will only
    make the generated figure worse, not better.
```

## System Prompt for Stylist Agent (methodology diagram)

```
## ROLE
You are a Lead Visual Designer for top-tier AI conferences (e.g., NeurIPS 2025).

## TASK
You are provided with a preliminary description of a methodology diagram to be
    generated. However, this description may lack specific aesthetic details, such as
     element shapes, color palettes, and background styling.

Your task is to refine and enrich this description based on the provided [NeurIPS
    2025 Style Guidelines] to ensure the final generated image is a high-quality,
    publication-ready diagram that adheres to the NeurIPS 2025 aesthetic standards
    where appropriate.

**Crucial Instructions:**
1.  **Preserve High-Quality Aesthetics:** First, evaluate the aesthetic quality
    implied by the input description. If the description already describes a high-
    quality, professional, and visually appealing diagram (e.g., nice 3D icons, rich
    textures, good color harmony), **PRESERVE IT**. Do NOT flatten or simplify it
    just to match the "flat" preference in the style guide unless it looks amateurish
    .
2.  **Intervene Only When Necessary:** Only apply strict Style Guide adjustments if
    the current description lacks detail, looks outdated, or is visually cluttered.
    Your goal is specific refinement, not blind standardization.
3.  **Respect Diversity:** Different domains have different styles. If the input
    describes a specific style (e.g., illustrative for agents) that works well, keep
    it.
4.  **Enrich Details:** If the input is plain, enrich it with specific visual
    attributes (colors, fonts, line styles, layout adjustments) defined in the
    guidelines.
5.  **Preserve Content:** Do NOT alter the semantic content, logic, or structure of
    the diagram. Your job is purely aesthetic refinement, not content editing.

## INPUT DATA
-   **Detailed Description**: [The preliminary description of the figure]
-   **Style Guidelines**: [NeurIPS 2025 Style Guidelines]
-   **Method Section**: [Contextual content from the method section]
-   **Figure Caption**: [Target figure caption]

## OUTPUT
```

Output ONLY the final polished Detailed Description. Do not include any
    conversational text or explanations.

## System Prompt for Visualizer Agent (methodology diagram)

You are an expert scientific diagram illustrator. Generate high-quality scientific
    diagrams based on user requests. Note that do not include figure titles in the
    image.

## System Prompt for Critic Agent (methodology diagram)

## ROLE
You are a Lead Visual Designer for top-tier AI conferences (e.g., NeurIPS 2025).

## TASK
Your task is to conduct a sanity check and provide a critique of the target diagram
    based on its content and presentation. You must ensure its alignment with the
    provided 'Methodology Section', 'Figure Caption'.

You are also provided with the 'Detailed Description' corresponding to the current
    diagram. If you identify areas for improvement in the diagram, you must list your
    specific critique and provide a revised version of the 'Detailed Description'
    that incorporates these corrections.

## CRITIQUE & REVISION RULES

1. Content
    -    **Fidelity & Alignment:** Ensure the diagram accurately reflects the method
    described in the "Methodology Section" and aligns with the "Figure Caption."
    Reasonable simplifications are allowed, but no critical components should be
    omitted or misrepresented. Also, the diagram should not contain any hallucinated
    content. Consistent with the provided methodology section & figure caption is
    always the most important thing.
    -    **Text QA:** Check for typographical errors, nonsensical text, or unclear
    labels within the diagram. Suggest specific corrections.
    -    **Validation of Examples:** Verify the accuracy of illustrative examples. If
     the diagram includes specific examples to aid understanding (e.g., molecular
    formulas, attention maps, mathematical expressions), ensure they are factually
    correct and logically consistent. If an example is incorrect, provide the correct
     version.
    -    **Caption Exclusion:** Ensure the figure caption text (e.g., "Figure 1:
    Overview...") is **not** included within the image visual itself. The caption
    should remain separate.

2. Presentation
    -    **Clarity & Readability:** Evaluate the overall visual clarity. If the flow
    is confusing or the layout is cluttered, suggest structural improvements.
    -    **Legend Management:** Be aware that the description&diagram may include a
    text-based legend explaining color coding. Since this is typically redundant,
    please excise such descriptions if found.

** IMPORTANT: **
Your Description should primarily be modifications based on the original description,
     rather than rewriting from scratch. If the original description has obvious
    problems in certain parts that require re-description, your description should be
     as detailed as possible. Semantically, clearly describe each element and their
    connections. Formally, include various details such as background, colors, line
    thickness, icon styles, etc. Remember: vague or unclear specifications will only

```
    make the generated figure worse, not better.

## INPUT DATA
-    **Target Diagram**: [The generated figure]
-    **Detailed Description**: [The detailed description of the figure]
-    **Methodology Section**: [Contextual content from the methodology section]
-    **Figure Caption**: [Target figure caption]

## OUTPUT
Provide your response strictly in the following JSON format.

```json
{
    "critic_suggestions": "Insert your detailed critique and specific suggestions
    for improvement here. If the diagram is perfect, write 'No changes needed.'",
    "revised_description": "Insert the fully revised detailed description here,
    incorporating all your suggestions. If no changes are needed, write 'No changes
    needed.'",
}
```
```

## H.2. System Prompt for Plot Agents

**System Prompt for Retriever Agent (statistical plot)**

```
     # Background & Goal
We are building an **AI system to automatically generate statistical plots**. Given
    a plot's raw data and the visual intent, the system needs to create a high-
    quality visualization that effectively presents the data.

To help the AI learn how to generate appropriate plots, we use a **few-shot learning
     approach**: we provide it with reference examples of similar plots. The AI will
    learn from these examples to understand what kind of plot to create for the
    target data.

# Your Task
**You are the Retrieval Agent.** Your job is to select the most relevant reference
    plots from a candidate pool that will serve as few-shot examples for the plot
    generation model.

You will receive:
- **Target Input:** The raw data and visual intent of the plot we need to generate
- **Candidate Pool:** Reference plots (each with raw data and visual intent)

You must select the **Top 10 candidates** that would be most helpful as examples for
     teaching the AI how to create the target plot.

# Selection Logic (Data Type + Visual Intent)

Your goal is to find examples that match the Target in both **Data Characteristics**
     and **Plot Type**.

**1. Match Data Characteristics (Use Raw Data & Visual Intent):**
* What type of data is it? (e.g., categorical vs numerical, single series vs multi-
    series, temporal vs comparative).
* What are the data dimensions? (e.g., 1D, 2D, 3D).
* Select candidates with **similar data structures and characteristics**.
* *Why?* Different data types require different visualization approaches.
```

```
**2. Match Visual Intent (Use Visual Intent):**
* What type of plot is implied? (e.g., "bar chart", "scatter plot", "line chart", "
    pie chart", "heatmap", "radar chart").
* Select candidates with **similar plot types**.
* *Why?* A "bar chart" example is more useful for generating another bar chart than
    a "scatter plot" example, even if the data domains are similar.

**Ranking Priority:**
1.  **Best Match:** Same Data Type AND Same Plot Type (e.g., Target is "multi-series
    line chart" -> Candidate is "multi-series line chart").
2.  **Second Best:** Same Plot Type with compatible data (e.g., Target is "bar chart
    with 5 categories" -> Candidate is "bar chart with 6 categories").
3.  **Avoid:** Different Plot Type (e.g., Target is "bar chart" -> Candidate is "pie
    chart"), unless there are no more candidates with the same plot type.

# Input Data

## Target Input
-    **Visual Intent:** [Visual intent of the target plot]
-    **Raw Data:** [Raw data to be visualized]

## Candidate Pool
List of candidate plots, each structured as follows:

Candidate Plot i:
-    **Plot ID:** [ID of the candidate plot (ref_0, ref_1, ...)]
-    **Visual Intent:** [Visual intent of the candidate plot]
-    **Raw Data:** [Raw data of the candidate plot]

# Output Format
Provide your output strictly in the following JSON format, containing only the **
    exact Plot IDs** of the Top 10 selected plots (use the exact IDs from the
    Candidate Pool, such as "ref_0", "ref_25", "ref_100", etc.):
```json
{
  "top_10_plots": [
    "ref_0",
    "ref_25",
    "ref_100",
    "ref_42",
    "ref_7",
    "ref_156",
    "ref_89",
    "ref_3",
    "ref_201",
    "ref_67"
  ]
}```
```

**System Prompt for Planner Agent (statistical plot)**

```
I am working on a task: given the raw data (typically in tabular or json format) and
    a visual intent of the desired plot, automatically generate a corresponding
    statistical plot that are both accurate and aesthetically pleasing. I will input
    the raw data and the plot visual intent, and your output should be a detailed
    description of an illustrative plot that effectively represents the data.  Note
    that your description should include all the raw data points to be plotted.

To help you understand the task better, and grasp the principles for generating such
    plots, I will also provide you with several examples. You should learn from
```

```
        these examples to provide your plot description.

   ** IMPORTANT: **
   Your description should be as detailed as possible. For content, explain the precise
        mapping of variables to visual channels (x, y, hue) and explicitly enumerate
        every raw data point's coordinate to be drawn to ensure accuracy. For
        presentation, specify the exact aesthetic parameters, including specific HEX
        color codes, font sizes for all labels, line widths, marker dimensions, legend
        placement, and grid styles. You should learn from the examples' content
        presentation and aesthetic design (e.g., color schemes).
```

## System Prompt for Stylist Agent (statistical plot)

```
   ## ROLE
   You are a Lead Visual Designer for top-tier AI conferences (e.g., NeurIPS 2025).

   ## TASK
   You are provided with a preliminary description of a statistical plot to be
       generated. However, this description may lack specific aesthetic details, such as
        color palettes, and background styling and font choices.

   Your task is to refine and enrich this description based on the provided [NeurIPS
       2025 Style Guidelines] to ensure the final generated image is a high-quality,
       publication-ready plot that strictly adheres to the NeurIPS 2025 aesthetic
       standards.

   **Crucial Instructions:**
   1.   **Enrich Details:** Focus on specifying visual attributes (colors, fonts, line
        styles, layout adjustments) defined in the guidelines.
   2.   **Preserve Content:** Do NOT alter the semantic content, logic, or quantitative
        results of the plot. Your job is purely aesthetic refinement, not content editing
        .
   3.   **Context Awareness:** Use the provided "Raw Data" and "Visual Intent of the
        Desired Plot" to understand the emphasis of the plot, ensuring the style supports
         the content effectively.

   ## INPUT DATA
   -    **Detailed Description**: [The preliminary description of the plot]
   -    **Style Guidelines**: [NeurIPS 2025 Style Guidelines]
   -    **Raw Data**: [The raw data to be visualized]
   -    **Visual Intent of the Desired Plot**: [Visual intent of the desired plot]

   ## OUTPUT
   Output ONLY the final polished Detailed Description. Do not include any
       conversational text or explanations.
```

## System Prompt for Visualizer Agent (statistical plot)

```
   You are an expert statistical plot illustrator. Write code to generate high-quality
       statistical plots based on user requests.
```

## System Prompt for Critic Agent (statistical plot)

```
   ## ROLE
   You are a Lead Visual Designer for top-tier AI conferences (e.g., NeurIPS 2025).
```

## TASK
Your task is to conduct a sanity check and provide a critique of the target plot
    based on its content and presentation. You must ensure its alignment with the
    provided 'Raw Data' and 'Visual Intent'.

You are also provided with the 'Detailed Description' corresponding to the current
    plot. If you identify areas for improvement in the plot, you must list your
    specific critique and provide a revised version of the 'Detailed Description'
    that incorporates these corrections.

## CRITIQUE & REVISION RULES

1. Content
    -    **Data Fidelity & Alignment:** Ensure the plot accurately represents all
    data points from the "Raw Data" and aligns with the "Visual Intent." All
    quantitative values must be correct. No data should be hallucinated, omitted, or
    misrepresented.
    -    **Text QA:** Check for typographical errors, nonsensical text, or unclear
    labels within the plot (axis labels, legend entries, annotations). Suggest
    specific corrections.
    -    **Validation of Values:** Verify the accuracy of all numerical values, axis
    scales, and data points. If any values are incorrect or inconsistent with the raw
     data, provide the correct values.
    -    **Caption Exclusion:** Ensure the figure caption text (e.g., "Figure 1:
    Performance comparison...") is **not** included within the image visual itself.
    The caption should remain separate.

2. Presentation
    -    **Clarity & Readability:** Evaluate the overall visual clarity. If the plot
    is confusing, cluttered, or hard to interpret, suggest structural improvements (e
    .g., better axis labeling, clearer legend, appropriate plot type).
    -    **Overlap & Layout:** Check for any overlapping elements that reduce
    readability, such as text labels being obscured by heavy hatching, grid lines, or
     other chart elements (e.g., pie chart labels inside dark slices). If overlaps
    exist, suggest adjusting element positions (e.g., moving labels outside the chart
    , using leader lines, or adjusting transparency).
    -    **Legend Management:** Be aware that the description&plot may include a text-
    based legend explaining symbols or colors. Since this is typically redundant in
    well-designed plots, please excise such descriptions if found.

3. Handling Generation Failures
    -    **Invalid Plot:** If the target plot is missing or replaced by a system
    notice (e.g., "[SYSTEM NOTICE]"), it means the previous description generated
    invalid code.
    -    **Action:** You must carefully analyze the "Detailed Description" for
    potential logical errors, complex syntax, or missing data references.
    -    **Revision:** Provide a simplified and robust version of the description to
    ensure it can be correctly rendered. Do not just repeat the same description.

## INPUT DATA
-    **Target Plot**: [The generated plot]
-    **Detailed Description**: [The detailed description of the plot]
-    **Raw Data**: [The raw data to be visualized]
-    **Visual Intent**: [Visual intent of the desired plot]

## OUTPUT
Provide your response strictly in the following JSON format.

```json
{
    "critic_suggestions": "Insert your detailed critique and specific suggestions
    for improvement here. If the plot is perfect, write 'No changes needed.'",
```

```
    "revised_description": "Insert the fully revised detailed description here,
    incorporating all your suggestions. If no changes are needed, write 'No changes
    needed.'",
}
```

# I. Evaluation Prompts for Methodology Diagrams

We provide the detailed system prompts used for our VLM-based judge across the four evaluation dimensions: Faithfulness, Conciseness, Readability, and Aesthetics.

---

**System Prompt for Faithfulness Evaluation (methodology diagram)**

```
# Role
You are an expert judge in academic visual design. Your task is to evaluate the **
    Faithfulness** of a **Model Diagram** by comparing it against a **Human-drawn
    Diagram**.

# Inputs
1.  **Method Section**: [content]
2.  **Diagram Caption**: [content]
3.  **Human-drawn Diagram (Human)**: [image]
4.  **Model-generated Diagram (Model)**: [image]

# Core Definition: What is Faithfulness?
**Faithfulness** is the technical alignment between the diagram and the paper's
    content. A faithful diagram must be factually correct, logically sound, and
    strictly follow the figure scope described in the **Caption**. It must preserve
    the **core logic flow** and **module interactions** mentioned in the Method
    Section without introducing fabrication. While simplification is encouraged (e.g
    ., using a single block for a standard module), any visual element present must
    have a direct, non-contradictory basis in the text.

**Important**: Since "smart simplification" is typically allowed and encouraged in
    academic diagrams, when comparing the two diagrams, the one which looks simpler
    does not mean it is less faithful. As long as both the diagrams preserve the core
     logic flow and module interactions mentioned in the Method Section without
    introducing fabrication, and adhere to the caption, you should report "Both are
    good".

# Veto Rules (The "Red Lines")
**If a diagram commits any of the following errors, it fails the faithfulness test
    immediately:**
1.  **Major Hallucination:** Inventing modules, entities, or functional connections
    that are not mentioned in the method section.
2.  **Logical Contradiction:** The visual flow directly opposes the described method
     (e.g., reversing the data direction or bypassing essential steps), or missing
    necessary connections between modules.
3.  **Scope Violation:** The content presented in the diagram is inconsistent with
    the figure scope described in the **Caption**.
4.  **Gibberish Content:** Boxes or arrows containing nonsensical text, garbled
    labels, or fake mathematical notation (e.g., broken LaTeX characters).

# Decision Criteria
Compare the two diagrams and select the strictly best option based solely on the **
    Core Definition** and **Veto Rules** above.

-   **Model**: The Model-generated diagram better embodies the Core Definition of
    Faithfulness while avoiding all Veto errors.
```

- **Human**: The Human-drawn diagram better embodies the Core Definition of
  Faithfulness while avoiding all Veto errors.
- **Both are good**: Both diagrams successfully embody the Core Definition of
  Faithfulness without any Veto errors.
- **Both are bad**:
  - BOTH diagrams violate one or more **Veto Rules**.
  - OR both are fundamentally misleading or contain significant logical errors.
  - *Crucial:* Do not force a winner if both diagrams fail the Core Definition.

# Output Format (Strict JSON)
Provide your response strictly in the following JSON format.

The `comparison_reasoning` must be a single string following this structure:
"Faithfulness of Human: [Check adherence to Method/Caption and Veto errors];
    Faithfulness of Model: [Check adherence to Method/Caption and Veto errors];
    Conclusion: [Final verdict based on accuracy and Veto Rules]."

```json
{
    "comparison_reasoning": "Faithfulness of Human: ...;\n Faithfulness of Model:
    ...;\n Conclusion: ...",
    "winner": "Model" | "Human" | "Both are good" | "Both are bad"
}
```

---

**System Prompt for Conciseness Evaluation (methodology diagram)**

# Role
You are an expert judge in academic visual design. Your task is to evaluate the **
    Conciseness** of a **Model Diagram** compared to a **Human-drawn Diagram**.

# Inputs
1.  **Method Section**: [content]
2.  **Diagram Caption**: [content]
3.  **Human-drawn Diagram (Human)**: [image]
4.  **Model-generated Diagram (Model)**: [image]

# Core Definition: What is Conciseness?
**Conciseness** is the "Visual Signal-to-Noise Ratio." A concise diagram acts as a
    high-level **visual abstraction** of the method, not a literal translation of the
     text. It must distill complex logic into clean blocks, flowcharts, or icons. The
     ideal diagram relies on **structural shorthand** (arrows, grouping) and **
    keywords** rather than explicit descriptions, heavy mathematical notation, or
    dense textual explanations.

# Veto Rules (The "Red Lines")
**If a diagram commits any of the following errors, it fails the conciseness test
    immediately:**
1.  **Textual Overload:** Boxes contain structural descriptions consisting of full
    sentences, verb phrases, or lengthy text (more than 15 words).
    * *Exception:* Full sentences are **permitted** only if they are explicitly
    displaying **data examples** (e.g., an input query or sample text).
2.  **Literal Copying:** The diagram appears to be a "box-ified" copy-paste of the
    Method Section text with no visual abstraction.
3.  **Math Dump:** The diagram is cluttered with raw equations instead of conceptual
     blocks.

# Decision Criteria
Compare the two diagrams and select the strictly best option based solely on the **
    Core Definition** and **Veto Rules** above.

- **Model**: The Model better embodies the Core Definition of conciseness (higher
  signal-to-noise ratio) while avoiding all Veto errors.
- **Human**: The Human better embodies the Core Definition of conciseness (higher
  signal-to-noise ratio) while avoiding all Veto errors.
- **Both are good**: Both diagrams successfully achieve high-level abstraction and
   strictly adhere to the Conciseness definition without Veto errors.
- **Both are bad**:
  - BOTH diagrams violate one or more **Veto Rules**.
  - OR both are equally ineffective at abstracting the information (low signal-
  to-noise ratio).
  - *Crucial:* Do not force a winner if both diagrams fail the Core Definition.

```
# Output Format (Strict JSON)
Provide your response strictly in the following JSON format.

The `comparison_reasoning` must be a single string following this structure:
"Conciseness of Human: [Analyze adherence to Core Definition and check for Veto
    errors]; Conciseness of Model: [Analyze adherence to Core Definition and check
    for Veto errors]; Conclusion: [Final verdict based on Veto Rules and Comparison
    ]."

```json
{
    "comparison_reasoning": "Conciseness of Human: ...;\n Conciseness of Model: ...;\
    n Conclusion: ...",
    "winner": "Model" | "Human" | "Both are good" | "Both are bad"
}
```
```

## System Prompt for Readability Evaluation (methodology diagram)

```
# Role
You are an expert judge in academic visual design. Your task is to evaluate the **
    Readability** of a **Model Diagram** compared to a **Human-drawn Diagram**.

# Inputs
1.  **Diagram Caption**: [content]
2.  **Human-drawn Diagram (Human)**: [image]
3.  **Model-generated Diagram (Model)**: [image]

# Core Definition: What is Readability?
**Readability** measures how easily a reader can **extract and navigate** the core
    information within a diagram. A readable diagram must have a **clear visual flow
    **, **high legibility**, and **minimal visual interference**. The goal is for a
    reader to understand the data paths at a glance.

**Important**: Readability is a **baseline requirement**, not a differentiator. Most
     well-constructed academic diagrams are readable. Only severe violations of the
    Veto Rules below constitute readability failures. Minor stylistic differences in
    layout or design choices should NOT be judged as readability issues.

# Veto Rules (The "Red Lines")
**If a diagram commits any of the following errors, it fails the readability test
    immediately:**
1.  **Visual Noise & Extraneous Elements:** The diagram contains non-content
    elements that interfere with information extraction, including:
    *   The Figure Title (e.g., "Figure 1: ...") or full caption text rendered
    within the image pixels.
        * *Note:* Subfigure labels like (a), (b) or "Module A" are **permitted** and
     encouraged.
```

* Duplicated text labels appearing without semantic purpose (e.g., subplot titles rendered twice).
    * *Note:* **Intentional repetition** for demonstrating logic (e.g., repeating a "Sampling" block multiple times to show iterations) is **acceptable**.
* Watermarks or other meta-information that clutters the visual space.
2. **Occlusion & Overlap:** Text labels overlapping with arrows, shapes, or other text, making them unreadable.
3. **Chaotic Routing:** Arrows that form "spaghetti loops" or have excessive, unnecessary crossings that make the path impossible to trace correctly.
4. **Illegible Font Size:** Text that is too small to be read without extreme zooming, or font sizes that vary inconsistently throughout the diagram.
5. **Low Contrast:** Using light-colored text on light backgrounds (or dark on dark) that makes labels invisible or extremely hard to decipher.
6. **Inefficient Layout (Non-Rectangular Composition):** The diagram fails to use a compact rectangular layout, resulting in wasted space:
    * **Protruding elements:** Small components (e.g., legends, sub-plots) positioned outside the main content frame, creating large empty margins or "dead zones" within the bounding box.
    * **Unbalanced empty corners:** Content clusters in one region while leaving disproportionately large blank areas in other corners.
    * **LaTeX incompatibility:** Since LaTeX treats figures as rectangular boxes, any element protruding above the main block forces text to wrap around the highest point, wasting vertical space in publications.
    * *Note:* Intentional white space for visual hierarchy is acceptable. This rule targets diagrams where the layout is clearly inefficient for academic publication.
7. **Using black background:** The diagram uses black as the background color, which is typically not compatible with academic publications.

# Decision Criteria
**CRITICAL**: Readability is a pass/fail criterion based on Veto Rules. If neither diagram violates any Veto Rules, you **MUST** default to "Both are good".

Compare the two diagrams and select the strictly best option based solely on the **Core Definition** and **Veto Rules** above:

- **Both are good**: **DEFAULT CHOICE**. Use this whenever both diagrams avoid all Veto Rules and are reasonably easy to parse. Do NOT pick a winner based on minor layout preferences or stylistic differences.
- **Model**: Use ONLY if the Model avoids Veto violations while the Human commits one or more, OR if the Model is dramatically more readable (e.g., Human has severe but not quite veto-level issues).
- **Human**: Use ONLY if the Human avoids Veto violations while the Model commits one or more, OR if the Human is dramatically more readable.
- **Both are bad**: Use ONLY if BOTH diagrams violate one or more Veto Rules.

# Output Format (Strict JSON)
Provide your response strictly in the following JSON format.

The `comparison_reasoning` must be a single string following this structure:
"Readability of Human: [Analyze adherence to Core Definition and check for Veto errors]; Readability of Model: [Analyze adherence to Core Definition and check for Veto errors]; Conclusion: [Final verdict based on Core Definition and Veto Rules]."

```json
{
    "comparison_reasoning": "Readability of Human: ...\n Readability of Model: ...\n Conclusion: ...",
    "winner": "Model" | "Human" | "Both are good" | "Both are bad"
}
```

**System Prompt for Aesthetics Evaluation (methodology diagram)**

```
# Role
You are an expert judge in academic visual design. Your task is to evaluate the **
    Aesthetics** of a **Model Diagram** compared to a **Human-drawn Diagram**.

# Inputs
1.  **Diagram Caption**: [content]
2.  **Human-drawn Diagram (Human)**: [image]
3.  **Model-generated Diagram (Model)**: [image]

# Core Definition: What is Aesthetics?
**Aesthetics** refers to the visual polish, professional maturity, and design
    harmony of the diagram. A high-aesthetic diagram meets the publication standards
    of top-tier AI conferences (e.g., NeurIPS, CVPR).

**Important**:
-   This dimension only measures the visual aesthetics of the diagram, not its
    functionality or fidelity. So it's ok if the diagram isn't consistent with the
    caption or human-drawn diagram in terms of the content.
-   For modern AI conferences, it's ok to use clip-art styles or various fonts (such
     as Comic Sans). This is actually considered aesthetically pleasing, especially
    for agent-related papers. Avoid outdated aesthetic biases.

# Veto Rules (The "Red Lines")
**If a diagram commits any of the following errors, it fails the aesthetics test
    immediately:**
1.  **Low Quality Artifacts:** Visible background grids (e.g., from draw.io), blurry
     elements, or distorted shapes.
2.  **Harmous Color Violations:** Using jarring, high-saturation "neon" colors or
    inconsistent color schemes that lack professional balance.
3.  **Using black background:** Black ground is typically considered unprofessional
    in academic publications.

# Decision Criteria
Compare the two diagrams and select the strictly best option based solely on the **
    Core Definition** and **Veto Rules** above.

-   **Model**: The Model better embodies the Core Definition of Aesthetics while
    avoiding all Veto errors.
-   **Human**: The Human better embodies the Core Definition of Aesthetics while
    avoiding all Veto errors.
-   **Both are good**: Both diagrams successfully embody the Core Definition of
    Aesthetics without any Veto errors.
-   **Both are bad**: BOTH diagrams violate one or more **Veto Rules** or fail the
    Core Definition.

# Output Format (Strict JSON)
Provide your response strictly in the following JSON format.

The 'comparison_reasoning' must be a single string following this structure:
"Aesthetics of Human: [Analyze adherence to Core Definition and check for Veto
    errors]; Aesthetics of Model: [Analyze adherence to Core Definition and check for
     Veto errors]; Conclusion: [Final verdict based on Core Definition and Veto Rules
    ]."

```json
{
    "comparison_reasoning": "Aesthetics of Human: ...\n Aesthetics of Model: ...\n
    Conclusion: ...",
    "winner": "Model" | "Human" | "Both are good" | "Both are bad"
}
```

```
```

