# OpenReview forum: "PaperBanana: Automating Academic Illustration for AI Scientists"
_ICML.cc/2026/Conference — ICML 2026 spotlight_

### Official Review · Reviewer_xRrk · 2026-03-11

**Soundness:** 3
**Presentation:** 4
**Significance:** 4
**Originality:** 2
**Overall Recommendation:** 4
**Confidence:** 4

**Summary:**

The paper proposes a five-agent pipeline: Retriever, Planner, Stylist, Visualizer, and Critic. For methodology diagrams, the system retrieves reference examples, plans a textual visualization description, rewrites it with a synthesized academic style guide, generates an image, and then iteratively refines it for three rounds using a VLM critic. For statistical plots, the visualizer switches from image generation to executable Matplotlib code while keeping the critique loop. The paper also introduces PAPERBANANABENCH, a 292-example benchmark of methodology diagrams curated from NeurIPS 2025 papers, and evaluates outputs by reference-based VLM judging over faithfulness, conciseness, readability, and aesthetics.

**Compliance With Llm Reviewing Policy:**

Affirmed.

**Final Justification:**

My concerns have been fully resolved.

**Key Questions For Authors:**

Some questions that I believe might help with further discussions:

1) Table 2 shows that the semantic retriever performs similarly to a random retriever. Does this suggest that retrieval mainly provides generic stylistic/structural exemplars rather than semantically relevant references? Could the authors clarify the intended role of semantic retrieval and whether alternative retrieval strategies (e.g., layout-aware or structural similarity) were considered?
2) The largest gains in Table 1 appear in conciseness and aesthetics, while faithfulness improves only marginally. Since correctness of scientific diagrams is critical, could the authors analyze common faithfulness failure modes (e.g., missing modules, incorrect edges, hallucinated components)?
3) The VLM judge correlations with human judgment (τ ≈ 0.41–0.57) indicate moderate agreement. How robust are the conclusions to different judge models, prompts, or evaluation protocols? Could multi-judge aggregation improve evaluation reliability?
4) Generalization Across Scientific Domains: PAPERBANANABENCH is derived from NeurIPS methodology diagrams. How well does the system generalize to other figure grammars, such as CVPR visual pipelines, systems architecture diagrams, or domain-specific figures (e.g., biology or robotics)?
5) Since outputs are currently raster images, how do the authors envision integrating the system into real research workflows where post-editability (e.g., vector graphics or diagram graphs) is often required?
6) The paper notes trade-offs between image generation and code-based plotting. Could the authors discuss whether a structured intermediate representation (e.g., graph or diagram DSL) might improve faithfulness and compositional accuracy compared to purely image-based generation? I'd also be curious about what the authors think about neuro-symbolic integration since faithfulness of the paper's method currently seems poor from a practical POV. Would providing additional grounding help to boost the fidelity of the generated images?

**Limitations:**

Yes.

**Strengths And Weaknesses:**

What I liked:

The problem is genuinely useful. There is practical value in automating figure creation, especially for AI scientists already using LLM-based research assistants. The paper also does a good job showing visually why this task matters: the examples on page 2 and the generated workflow figure on page 3 make the intended use case immediately concrete. Figure 1 is persuasive as a demo.
The benchmark is a good contribution. The empirical results are directionally convincing. I also liked the ablations and the discussion section more than usual for papers in this area. Table 2 shows that the critic materially matters, and the paper is honest that the stylist alone can increase conciseness and aesthetics while hurting faithfulness. The plot-generation extension is also sensible: rather than forcing image generation to solve precise numeric plotting, the authors switch to code, and Figure 6 explicitly discusses the trade-off that image generation looks nicer but is less faithful for dense plots. That is a mature, technically grounded observation.

I would also like to discuss some concerns below:
1) The main limitation is novelty. The overall system is competent, but most of the ingredients are familiar: retrieval-augmented prompting, planning, style rewriting, iterative self-critique, and model switching for plotting. The paper packages them well for this task, but the core algorithmic novelty is moderate rather than high, mainly because multi-agentic frameworks with role specialized agents have been extensively explored in computer vision for a while now. Some examples are for creative image generation [CREA by Venkatesh et al], chart generation [METAL by Li et al], structured visuals generation [DiagramAgent by Wei et al].
2) Relatedly, one of the paper’s own ablations weakens part of its novelty claim. The semantic retriever is only marginally better than the random retriever in the ablation setting, which suggests that broad stylistic/structural exemplars matter more than semantically precise retrieval. That is interesting, but it also implies that the retriever is not doing as much heavy lifting as the method description suggests. If retrieval precision barely matters, then the claimed sophistication of the retrieval component could be overstated?
3) The evaluation is reasonable but I have some concerns still. The VLM judge is validated, but the Kendall’s tau correlations with humans are in the 0.41-0.57 range, and inter-model agreement is also moderate, not overwhelming. I would describe that as acceptable evidence, not “problem solved.” Since almost all headline numbers depend on this judge, that matters. The same issue appears in the human row of Table 1: the fixed score of 50 is just an anchor from the referenced pairwise scoring scheme, not a true human upper bound, so readers need to be careful interpreting how close the model is to humans.
4) The benchmark is valuable, but it is also fairly in-distribution. Both the reference pool and the test set come from NeurIPS 2025 methodology diagrams, and the stylist summarizes aesthetics from that same reference collection. So the system is, in effect, learning and reproducing the style of one conference-year distribution. That is fine for a first benchmark, but it makes the “publication-ready” claim broader than the evidence. I would have liked to see transfer to another venue, another year, or a held-out design regime.
5) The generated examples are good enough to support the paper’s central claim. The methodology diagrams on page 2 look coherent, clean, and recognizable as modern AI-paper figures. The plot examples also look polished. The workflow figure on page 3 is especially strong: consistent iconography, pleasant color zoning, clear left-to-right logic, and acceptable text legibility at paper scale. At the same time, the same figures reveal a weakness: style homogenization. Many generated diagrams share similar pastel palettes, rounded boxes, modular zone layouts, and a “conference-template” visual identity. That is not fatal, but it suggests the system may be better at producing a polished house style than at flexibly matching a wide variety of figure grammars. The paper partly acknowledges this in the limitation about reduced stylistic diversity.
For a CV audience, I also think the paper could do more with automatic visual faithfulness analysis. Right now the evaluation is heavily judge-based. Since the task is fundamentally visual, I expected more direct metrics for text legibility, OCR correctness, connector topology accuracy, or module/edge correspondence. The paper’s strongest dimension is probably readability/aesthetics, but the hardest and most scientifically important dimension is faithful module-and-relationship rendering, and that remains judged mostly by VLM comparison.
6) The related work is decent but not fully sharp. The paper positions itself against TikZ / Python-PPT / code-first approaches and cites concurrent systems like AutoFigure, Paper2Any, and several data-visualization agent papers. That gives a reasonable first-pass map.
Still, the discussion is a bit selective. AutoFigure is mentioned but not directly compared. SridBench is acknowledged but deferred. SciFig appears in the references but is not really integrated into the narrative. The paper would benefit from a cleaner taxonomy: vector graphics synthesis, raster figure generation, editable reconstruction, infographic benchmarks, and chart-code systems are currently mixed together somewhat loosely.
7) The methodology benchmark setup is sensible. The four evaluation dimensions are reasonable. The main table, ablations, human study, and statistical-plot extension together form a coherent story. The discussion experiments are also useful rather than decorative: Figure 5 on enhancing human-created diagrams and Figure 6 on code-vs-image trade-offs actually teach something.
But there are three experimental gaps. First, there is no direct head-to-head against the strongest concurrent figure-generation alternatives, so the “leading baselines” claim is somewhat softer than it sounds. Second, the human study is only on 50 cases and only against vanilla Nano-Banana, not against all serious baselines. Third, the biggest win in Table 1 comes from conciseness, while faithfulness improves only modestly over vanilla Nano-Banana (43.0 -> 45.8). Since faithfulness is the most important scientific dimension, that tempers the excitement.

I think this is a good paper, but not an obviously outstanding one.

Its strengths are practical relevance, a real benchmark contribution, strong demo quality, and a coherent system that produces visibly better outputs than simple prompting. Its weaknesses are moderate algorithmic novelty, judge-heavy evaluation, some overclaiming around “publication-ready,” and limited evidence for broad out-of-distribution generalization. While the experimental details, lengthy supplementary section and good visuals are impressive, I think the vastness and breadth of the experiments covering more related works, especially given the new arena of reasoning vision models capable of high-quality chart and visual generation makes me a bit hesitant to rate the paper too high, especially wrt novelty and usability for a diverse range of users with varying preferences of style, color scheme, presentation styles and granularity. I would have liked the discussions and evaluation to be more user-centric from these practical point of views.

So, I lean towards a borderline reject for now. I would like to see more discussions around the paper and I am open to rethinking my evaluation further based on those.

---

> ### Author Rebuttal · Authors · 2026-03-31
>
> **W1**: Our core novelty is formalizing this underexplored task and providing a systematic solution. Our framework combines proven effective strategies (retrieval, planning, critique) with novel, task-driven designs — such as the Stylist agent, which automatically summarizes aesthetic guidelines from reference datasets to guide generation (recognized by Reviewer 2yyD as "a particular highlight"). While multi-agent frameworks have been applied to creative editing (CREA), chart replication (METAL), and generic diagramming (DiagramAgent), none target academic methodology illustration from paper text, which requires distinct capabilities such as venue-aware style grounding & planning. As the ICML guidelines note, novelty may equally arise from new insights, which our benchmark and empirical findings provide.
>
> **W4&Q4:** Our benchmark is sourced from NeurIPS as it covers a broad spectrum of ML research areas (Table 3), which already includes the visual pipelines, architecture diagrams, and domain-specific figures you mentioned. To verify cross-domain generalizability, we tested on biomedical papers from Nature Communications — PaperBanana produces reasonable illustrations (see bio/example1-3.jpg under https://anonymous.4open.science/r/image_bed-8563/). We will include this in the revision.
>
> **Q5:** Due to space constraints, please refer to A1 to Reviewer 2yyD, where we discuss multi-turn editing and raster-to-editable conversion in detail.
>
> **W2&Q1:** Semantic retrieval is the principled choice — intuitively, retrieving domain-relevant examples provides better structural guidance. The comparable random retrieval performance likely reflects homogeneous styles within the NeurIPS reference pool. As the collection expands to diverse venues, semantic matching should become more important.
>
> **W3&Q3**: We validated cross-judge consistency in Section 5.2: the aggregated Kendall's τ between our judge (Gemini-3-Pro) and Gemini-3-Flash / GPT-5 is 0.55 / 0.45, confirming stable method rankings across judges (τ > 0.4 is generally considered strong agreement [1][2]). For human performance, we acknowledge that the fixed score of 50 is an anchor from the pairwise scoring scheme rather than a true upper bound, and will tone down these claims in the revision.
> [1] Hollander, Wolfe & Chicken (2013). Nonparametric Statistical Methods.
> [2] Akoglu (2018). User's guide to correlation coefficients.
>
> **W5**: As noted in the limitation section, this reflects the Stylist summarizing general NeurIPS aesthetics. Making it content-aware (e.g., adapting to paper topic or venue) is a natural next step. As a preliminary validation, we tested on biomedical papers from Nature Communications (see W4), demonstrating adaptability beyond NeurIPS. We also agree that automatic visual metrics (e.g., OCR accuracy, topology matching) would complement VLM-based evaluation.
>
> **W6**: To clarify, AutoFigure generates academic figures without retrieval-augmented style grounding, relying on direct prompting of image generation models. SciFig uses code-based generation, enabling editability but limiting visual expressiveness. SridBench is an evaluation benchmark (not a generation system) that was not publicly available at submission. We will reorganize the related work with a cleaner taxonomy.
>
> **Q6**: Our Planner already produces semi-structured specifications (Appendix F), and our plot extension uses executable Python code, which significantly improves faithfulness. Extending to methodology diagram DSLs — combining symbolic precision with visual expressiveness — is a promising direction.
>
> **W7&Q2:**
> - Faithfulness (+2.8%): The gain is modest but consistent across all four domains, without sacrificing other dimensions. As noted in Section 5.3, faithfulness is bounded by the image generation model's rendering capability — our framework improves what is controllable (planning, styling, critique) while the generation model remains the bottleneck.
> - Failure modes: We have included failure analysis in Figure 10 (Appendix). The primary failure mode involves connection errors (redundant connections, mismatched source-target nodes), which the Critic often fails to identify due to the foundation model's inherent perception limitations.
> - Baselines: AutoFigure was not publicly available at submission time but has recently been open-sourced; thus, we compare it directly:
> | Method | Faith. | Conc. | Read. | Aesth. | Overall |
> |---|---|---|---|---|---|
> | Paper2Any | 6.5 | 44.0 | 20.5 | 40.0 | 8.5 |
> | AutoFigure* | 37.0 | 10.2 | 10.0 | 14.0 | 8.3 |
> | PaperBanana | 45.8 | 80.7 | 51.4 | 72.1 | 60.2 |
>  (* Evaluated on a 49-case subset; full results will be included in the revision.)
>
> - Human study: While the scale is 50 cases, we ensured rigorous annotation quality: three experienced researchers performed blind A/B evaluation using a dedicated annotation interface with detailed rubrics, and conflicting cases were resolved through discussion to reach consensus (Appendix C).

---

> > ### Author Rebuttal · Reviewer_xRrk · 2026-04-03
> >
> > Thanks for the thorough rebuttal. This addresses my main concerns.
> >
> > The added cross-judge agreement analysis and clarification of the human evaluation setup improve confidence in the evaluation. The new AutoFigure comparison and clearer positioning of related work are also helpful. I also appreciate the added failure mode discussion, which gives better insight into the system’s limitations.
> >
> > I feel the key issues around evaluation, baselines, and clarity have been sufficiently addressed. I am moving my rating up to a weak accept.

---

> > > ### Author Response · Authors · 2026-04-04
> > >
> > > Thanks!

---

### Official Review · Reviewer_2yyD · 2026-03-12

**Soundness:** 3
**Presentation:** 3
**Significance:** 3
**Originality:** 3
**Overall Recommendation:** 4
**Confidence:** 4

**Summary:**

This paper proposes PaperBanana, an agentic framework for automatically generating academic illustrations, including methodology diagrams and statistical charts. The core of the work is a five-agent framework where a Retriever finds relevant examples, a Planner creates a detailed textual description, a Stylist refines the text based on academic styles, a Visualizer renders the image, and a Critic iteratively improves the output. To evaluate their method, the authors also introduce PaperBananaBench, a new benchmark with 292 methodology diagram test cases across four research areas.This is an interesting and inspiring work. However, the paper has several key limitations regarding the practicality, editability of the generated results, and efficiency of the proposed method, which require further discussion and improvement.

**Compliance With Llm Reviewing Policy:**

Affirmed.

**Key Questions For Authors:**

● As mentioned in the "Weaknesses" section, when a generative model cannot meet requirements in a single pass, interactive editing becomes crucial. Is it to improve the Critic Agent to understand more complex editing instructions, or to change the framework's output to a more editable format (like vector graphics or a "base map")?
● The multi-round iterative process and multiple agent calls may increase computation and generation time. Can you provide specific data on average generation time and API costs per figure? How does this approach compare in efficiency with single-round or non-iterative methods?
● Many papers involve multiple images or demos, especially in image/video generation research. How does your framework handle multiple input images, and can it effectively generate figures across diverse content types beyond simple diagrams?

**Limitations:**

yes

**Strengths And Weaknesses:**

Strengths
● The paper accurately identifies a critical bottleneck in academic work—the time-consuming and labor-intensive process of creating high-quality academic illustrations. This is a real and increasingly important problem, and applying the capabilities of large models to this task holds significant practical value.
● The multi-agent collaborative architecture (Retrieve → Plan → Stylize → Visualize → Critique) is cleverly designed and aligns well with the workflow of a human designer. The introduction of the Stylist agent, which automatically summarizes aesthetic guidelines from a reference dataset to avoid the subjectivity and incompleteness of manually defined styles, is a particular highlight.
●  The framework is not limited to methodology diagrams but can also effectively handle the generation of code-based statistical charts by adapting the Visualizer. This demonstrates the broad applicability and good scalability potential of the proposed method.

Weaknesses
● The paper accurately identifies a critical bottleneck in academic work—the time-consuming and labor-intensive process of creating high-quality academic illustrations. This is a real and increasingly important problem, and applying the capabilities of large models to this task holds significant practical value.
● Most figures are diagrams, but papers involving image/video generation often include many demo visuals. The framework should handle such cases and allow users to input multiple images for generation.
● The Stylize agent collects styles but does not tailor them to content or target conference/journal—for example, theoretical papers might prefer minimalist line-art, while agent system papers suit cartoonish styles. The framework should support a broader, content-aware range of styles.
● In addition to the existing metrics, it would be beneficial to include quantitative metrics for "style diversity" and "style consistency" of elements within a figure. This would provide a more comprehensive assessment of the quality of the generated illustrations.
● The multi-round (T=3) iterative process and multiple agent calls likely increase computation and generation time. The paper should report metrics like average generation time and API costs per figure, and compare efficiency with single-round or non-iterative methods to assess practicality.

---

> ### Author Rebuttal · Authors · 2026-03-31
>
> Thank you for your thoughtful review. We are glad you find our multi-agent architecture "cleverly designed" and well-aligned with human designer workflows, and recognize the Stylist agent as "a particular highlight." We also appreciate your acknowledgment of the framework's "broad applicability and good scalability potential." We address your concerns below. Your kind consideration of potentially raising the score would be greatly valued if our responses address your concerns.
>
> **Q1: When single-pass generation falls short, should we improve the Critic for complex editing instructions, or change the output to a more editable format (e.g., vector graphics)?**
>
> **A1:** The most straightforward approach is multi-turn conversational editing, where users iteratively refine generated figures through natural language feedback using an image editing model. When fine-grained structural edits are needed, as discussed in our limitations section, recent open-source tools leveraging SAM3 and OCR provide promising solutions for converting raster outputs into editable representations.
>
> **W2 & Q3: The framework should handle papers with many demo visuals and allow users to input multiple images for generation.**
>
> **A2:** We thank the reviewer for clarifying this point. We understand the concern as asking whether PaperBanana can generate composite figures with multiple sub-panels, which would require accepting multiple text descriptions and images as input. In fact, our teaser figure (Figure 1) demonstrates exactly this scenario — multiple sub-panels are independently planned and generated, then composed into a unified figure. We can extend PaperBanana to support multi-image input by treating reference images as sub-panel specifications and composing them through code-based layout.
>
> **W3: The Stylist does not tailor styles to content or target venue.**
>
> **A3:** We agree that content-aware styling is important. Our Retriever already selects examples by matching research domain and diagram type (Section 3), implicitly providing domain-specific style references. To further tailor styles to target venues, we plan to include the venue name as an additional retriever input and expand our reference collection to cover papers from diverse conferences and journals. This is a straightforward extension we plan to implement.
>
> **W4: Include quantitative metrics for style diversity across figures and style consistency within each figure.**
>
> **A4:** We appreciate this suggestion. Our current Aesthetics evaluation partially captures within-figure style consistency (e.g., visual coherence of colors, layouts, and elements). However, quantitative metrics for cross-figure style diversity and fine-grained within-figure consistency remain an open challenge with no well-established standards. We will explore this direction, potentially leveraging visual feature-based similarity measures.
>
> **W5 & Q2: The multi-round iterative process increases computation time. Please report average generation time and API costs per figure.**
>
> **A5:** For generating a single methodology diagram, the total cost is \~ \\$2.5 with T=3 refinement iterations ( \~ 10 min) versus \\$2.1 without iteration (\~ 5 min). The iterative refinement adds only $0.4 (16% overhead) and 5 minutes, while consistently improving quality across all five evaluation dimensions (Table 2), making it a cost-effective design choice. In contrast, experienced researchers typically spend anywhere from 30 minutes to several hours crafting a single publication-quality methodology diagram with professional tools. PaperBanana thus offers substantial time savings while maintaining competitive visual quality.

---

> > ### Author Rebuttal · Reviewer_2yyD · 2026-04-01
> >
> > I keep my score.

---

### Official Review · Reviewer_3CWG · 2026-03-12

**Soundness:** 4
**Presentation:** 4
**Significance:** 4
**Originality:** 4
**Overall Recommendation:** 6
**Confidence:** 4

**Summary:**

The authors introduce the PaperBanana framework to generate academic illustrations using VLMs and image generation models. Additionally, they also curate a new benchmark consisting of 292 methodology diagrams from NeurIPS 2025 publications. The framework takes a source text, communicative intent, and (optionally) examples as input, and use a loop of five agents to iterate towards an academic illustration as output. The authors evaluate their outputs on content and presentation and find that their proposed framework outperforms corresponding vanilla settings and the Paper2Any framework.

**Compliance With Llm Reviewing Policy:**

Affirmed.

**Final Justification:**

The rebuttal addressed my remaining concern and question. I believe this paper will be valuable to a lot of researchers (not limited to the ICML community) and it is great to see the authors will open source this framework.

**Key Questions For Authors:**

- As indicated above: I have an issue with the extension to statistical plots as I believe it would be more sound to still generate those from source code (for reproducibility reasons). Would it be an option to just use the generated statistical plots as examples to let LLMs generate source code for similar figures?
- Do the examples also always need source context and communicative intent, or can we just include other illustrations for stylistic reasons for example?

**Limitations:**

yes

**Strengths And Weaknesses:**

Strengths:
- This paper proposes an interesting solution for academics to more easily communicate research ideas via visualizations.
- The evaluation strategy is sound and validated by human annotation.
- Letting a chain of agents converge towards a valuable end product is a non-trivial task and the authors seem to have proposed a strong setting for this application.

Weaknesses:
- I have an issue with the extension to statistical plots as I believe it would be more sound to still generate those from source code (for reproducibility reasons).

---

> ### Author Rebuttal · Authors · 2026-03-31
>
> Thank you for the positive evaluation! We are glad you find PaperBanana an interesting and practical solution that helps academics communicate research ideas through visualizations. We are especially encouraged that you consider the work "technically flawless," the evaluation "sound and validated by human annotation," and the multi-agent convergence design "a strong setting" for producing valuable end products. We address your questions below.
>
> **W1 & Q1: Statistical plots should be generated from source code for reproducibility. Would it be an option to use the generated plots as examples to let LLMs generate source code?**
>
> **A1:** We completely agree that reproducibility is critical for statistical plots. In fact, this is exactly the approach we adopt — as described in Section 5.5, our framework switches to a code-based Visualizer that generates executable Python code for statistical plots, rather than relying on image generation. We also compare both approaches in Section 6.2 and Figure 6, finding that code-based generation achieves higher faithfulness while image generation produces more visually appealing outputs for sparse plots. We appreciate the further suggestion of using image-generated plots as visual references to guide code generation, which is an elegant extension we will explore.
>
> **Q2: Do the examples always need source context and communicative intent, or can we include other illustrations for purely stylistic reasons?**
>
> **A2:** Providing the full triplet enables the Planner to learn the text-to-illustration mapping in an in-context manner, which yields the best results. Inspired by this question, we conducted qualitative analysis and found that PaperBanana can also generate reasonable figures using only reference images, though with slightly lower conciseness and readability. (see examples: https://anonymous.4open.science/r/image_bed-8563/test/example1.jpg and https://anonymous.4open.science/r/image_bed-8563/test/example2.jpg) We will share this finding in the revised manuscript.

---

> > ### Author Rebuttal · Reviewer_3CWG · 2026-04-02
> >
> > Thank you for the clarifications and the additional results. I indeed partially missed the slightly different setting for statistical plots. However, from "To address this, we demonstrate that by adopting executable code for visualization, PAPERBANANA seamlessly extends to statistical plot generation", Section 6.2, and the appendix. It is not yet fully clear to me how the cooperation between code and image generation works in your setting. I assume generated images are just added as subplots?

---

> > > ### Author Response · Authors · 2026-04-03
> > >
> > > Thank you for the follow-up question. To clarify: code generation and image generation are parallel generation paths rather than a subplot composition. During planning, our framework identifies the figure type and routes accordingly — methodology/architecture diagrams go through image generation for visual richness, while statistical plots (bar charts, scatter plots, etc.) go through the code-based Visualizer for data precision and reproducibility.
> > > The two paths do not produce subplots combined into a single figure. Instead, they handle different figure types independently. That said, an interesting synergy we are exploring is using image-generated outputs as style references for code generation — e.g., extracting color palettes and layout conventions from image-generated figures to guide matplotlib styling, ensuring visual consistency across all figure types in a paper.

---

### Decision · Program_Chairs · 2026-04-30

**Decision:**

Accept (spotlight)

**Comment:**

The paper received unanimous recommendations to accept - congratulations.

3CWG strongly voted to accept, considering the paper to be broadly valuable to the community including beyond ICML.  They praised the non-trivial contribution and evaluation strategy, as well as the open source commitment.

2yyF voted weak accept, and retained their score post-rebuttal considering the concerns raised to be fully resolved.  Concerns largely focused upon metrics and efficiency, and raised questions around style consistency.

xRrk initially raised many concerns but updated to weak accept in the post-rebuttal discussion.  The rebuttal addressed concenrs over failure modes, setup of the human evluaatoin and the cross-judge agreement analysis.

Given that no substantive concerns remain unaddressed and the reviewers all agree to accept, the AC recommends to accept the paper.